# Towards Trustworthy Reranking:
# A Simple yet Effective Abstention Mechanism

**Hippolyte Gisserot-Boukhlef**                    *hippolyte.gisserot-boukhlef@centralesupelec.fr*
*Artefact Research Center*
*MICS, CentraleSupélec, Université Paris-Saclay*

**Manuel Faysse**                                  *manuel.faysse@centralesupelec.fr*
*Illuin Technology*
*MICS, CentraleSupélec, Université Paris-Saclay*

**Emmanuel Malherbe**                              *emmanuel.malherbe@artefact.com*
*Artefact Research Center*

**Céline Hudelot**                                 *celine.hudelot@centralesupelec.fr*
*MICS, CentraleSupélec, Université Paris-Saclay*

**Pierre Colombo**                                 *pierre@equall.ai*
*Equall.ai*
*MICS, CentraleSupélec, Université Paris-Saclay*

**Reviewed on OpenReview:** *https://openreview.net/forum?id=iMKUMWfRIj*

## Abstract

Neural Information Retrieval (NIR) has significantly improved upon heuristic-based Information Retrieval (IR) systems. Yet, failures remain frequent, the models used often being unable to retrieve documents relevant to the user's query. We address this challenge by proposing a lightweight abstention mechanism tailored for real-world constraints, with particular emphasis placed on the reranking phase. We introduce a protocol for evaluating abstention strategies in black-box scenarios (typically encountered when relying on API services), demonstrating their efficacy, and propose a simple yet effective data-driven mechanism. We provide open-source code for experiment replication and abstention implementation, fostering wider adoption and application in diverse contexts.

## 1 Introduction

In recent years, NIR has emerged as a promising approach to addressing the challenges of IR on various tasks (Guo et al., 2016; Mitra & Craswell, 2017; Zhao et al., 2022). Central to the NIR paradigm are the pivotal stages of retrieval and reranking (Robertson et al., 1995; Ni et al., 2021; Neelakantan et al., 2022), which collectively play a fundamental role in shaping the performance and outcomes of IR systems (Thakur et al., 2021). While the objective of the retrieval stage is to efficiently fetch candidate documents from a vast corpus based on a user's query (frequentist (Ramos et al., 2003; Robertson et al., 2009) or bi-encoder-based approaches (Karpukhin et al., 2020)), reranking aims at reordering these retrieved documents in a slower albeit more effective way, using more sophisticated techniques (bi-encoder- or cross-encoder-based (Nogueira & Cho, 2019; Khattab & Zaharia, 2020)).[1]

However, despite advancements in NIR techniques, retrieval and reranking processes frequently fail, due for instance to the quality of the models being used, to ambiguous user queries or insufficient relevant documents

---

[1]The most common approach is to have a quick retriever that scales well and a more computationally intensive reranker that reorders the top retrieved contexts.

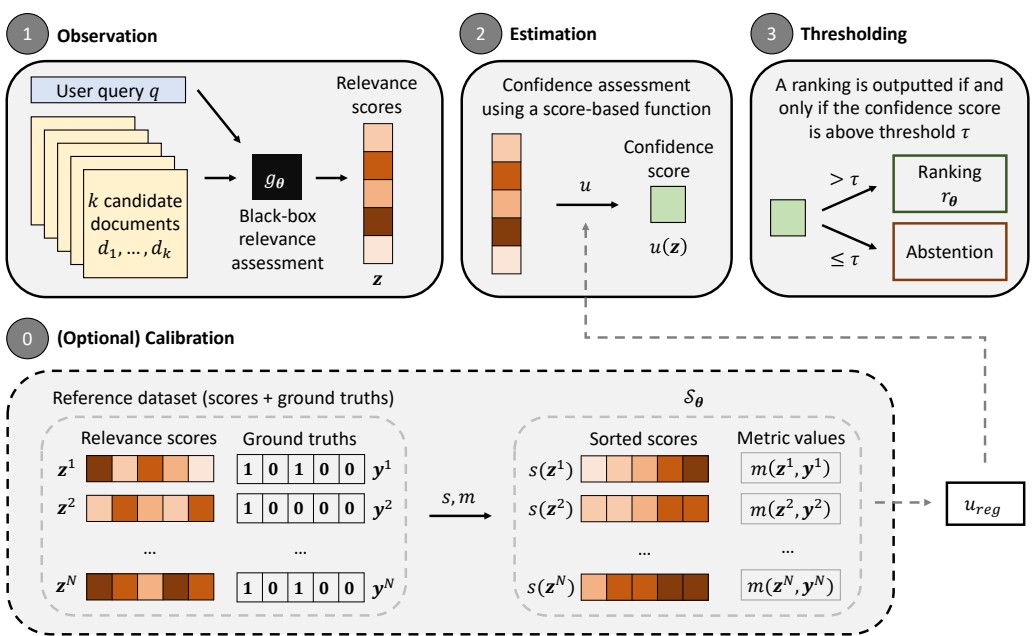

Figure 1: Procedure diagram for black-box confidence estimation and abstention decision in a reranking setting. In the reference-free scenario, confidence function $u$ is a simple heuristic (e.g., maximum). In the data-driven scenario, $u$ is a light non-trivial function of the relevance scores (e.g., learned linear combination).

in the database. These can have unsuitable consequences, particularly in contexts like Retrieval-Augmented Generation (RAG) (Lewis et al., 2020), where retrieved information is directly used in downstream tasks, thus undermining the overall system reliability (Yoran et al., 2023; Wang et al., 2023; Baek et al., 2023).

Recognizing the critical need to address these challenges, the implementation of abstention mechanisms emerges as a significant promise within the IR landscape. This technique offers a pragmatic approach by abstaining from delivering results when the model shows signs of uncertainty, thus ensuring users are not misled by potentially erroneous information. Although the subject of abstention in machine learning has long been of interest (Chow, 1957; El-Yaniv & Wiener, 2010; Geifman & El-Yaniv, 2017; 2019), most research has focused on the classification setting, leaving lack of significant initiatives in IR. A few related works propose computationally intensive learning strategies (Cheng et al., 2010; Mao et al., 2023) that are difficult to implement in the case of NIR, where models can have up to billions of parameters.

**Scope.** In this paper, we explore lightweight abstention mechanisms that adhere to realistic industrial constraints, namely **(i)** black-box access-only to query-documents relevance scores, which is typically the case when relying on API services, **(ii)** marginal computational and latency overhead, **(iii)** configurable abstention rates depending on the application. In line with industrial applications of IR systems for RAG or search engines, our method focuses on assessing the reranking quality of the top retrieved documents.

**Contributions.** Our key contributions are:

1. We devise a detailed protocol for evaluating abstention strategies in a black-box reranking setting and demonstrate their relevance.
2. We introduce a simple yet effective reference-based abstention mechanism, outperforming reference-free heuristics at zero incurred cost. We perform ablations and analyses on this method and are confident about its potential in practical settings.
3. We release a code package[2] and artifacts[3] to enable full replication of our experiments and the implementation of plug-and-play abstention mechanisms for any use case.

---

[2]https://github.com/artefactory/abstention-reranker, under MIT license.
[3]https://huggingface.co/collections/artefactory/abstention-reranking.

## 2 Problem Statement & Related Work

### 2.1 Notations

**General notations.** Let $\mathcal{V}$ denote the vocabulary and $\mathcal{V}^*$ the set of all possible textual inputs (Kleene closure of $\mathcal{V}$). Let $\mathcal{Q} \subset \mathcal{V}^*$ be the set of queries and $\mathcal{D} \subset \mathcal{V}^*$ the document database. In the reranking setting, a query $q \in \mathcal{Q}$ is associated with $k \in \mathbb{N}$ candidate documents $(d_1, \cdots, d_k) \in \mathcal{D}^k$ coming from the preceding retrieval phase, some of them being relevant to the query and the others not. When available, we denote by $\mathbf{y} \in \{0,1\}^k$ the ground truth vector, such that $y_j = 1$ if $d_j$ is relevant to $q$ and $y_j = 0$ otherwise.[4]

**Dataset.** When specified, we have access to a labeled reference dataset $\mathcal{S}$ composed of $N \in \mathbb{N}$ instances, each of them comprising of a query and $k$ candidate documents, $x = (q, d_1, \cdots, d_k) \in \mathcal{Q} \times \mathcal{D}^k$, and a ground truth $\mathbf{y} \in \{0,1\}^k$. Formally,

$$\mathcal{S} = \left\{ \left( x^i, \mathbf{y}^i \right) \right\}_{i=1}^{N} . \tag{1}$$

**Relevance scoring.** A crucial step in the reranking task is the calculation of query-document relevance scores. We denote by $f_{\boldsymbol{\theta}} : \mathcal{Q} \times \mathcal{D} \to \mathbb{R}$ the encoder-based relevance function that maps a query-document pair to its corresponding relevance score, with $\boldsymbol{\theta} \in \Theta$ standing for the encoder parameters. We also define $g_{\boldsymbol{\theta}} : \mathcal{Q} \times \mathcal{D}^k \to \mathbb{R}^k$, the function taking a query and $k$ candidate documents as input and returning the corresponding vector of relevance scores. Formally, given $x \in \mathcal{Q} \times \mathcal{D}^k$,

$$g_{\boldsymbol{\theta}} (x) = (f_{\boldsymbol{\theta}} (q, d_1), \cdots, f_{\boldsymbol{\theta}} (q, d_k)) . \tag{2}$$

Additionally, we define the sort function $s : \mathbb{R}^k \to \mathbb{R}^k$ that takes a vector of scores as input and returns its sorted version in increasing order. For a given $\mathbf{z} \in \mathbb{R}^k$, $s(\mathbf{z})$ is such that $s_1(\mathbf{z}) \leq \cdots \leq s_k(\mathbf{z})$.[5] We denote by $s_{\boldsymbol{\theta}} : \mathcal{Q} \times \mathcal{D}^k \to \mathbb{R}^k$ the function that takes a query and $k$ candidate documents and directly returns the sorted relevance scores, i.e.,

$$s_{\boldsymbol{\theta}} = s \circ g_{\boldsymbol{\theta}} \tag{3}$$

**Document ranking.** Once the scores are computed, we have a notion of the relevance level of each document with respect to the query. The idea is then to rank the documents by relevance score. For a given vector of scores $\mathbf{z} \in \mathbb{R}^k$, let $r(\mathbf{z}) \in \mathfrak{S}_k$ denote the corresponding vector of positions in the ascending sort, $s(\mathbf{z})$, where $\mathfrak{S}_k$ is the symmetric group of $k$ elements comprising of the $k!$ permutations of $\{1, \cdots, k\}$. In other terms, $r_i(\mathbf{z}) = j$ if $z_i$ is ranked $j^{\text{th}}$ in the ascending sort of $\mathbf{z}$. We then define the full reranking function $r_{\boldsymbol{\theta}} : \mathcal{Q} \times \mathcal{D}^k \to \mathfrak{S}_k$, taking a query and $k$ candidate documents as input and returning the corresponding ranking by relevance score. More formally,

$$r_{\boldsymbol{\theta}} = r \circ g_{\boldsymbol{\theta}} \tag{4}$$

**Instance-wise metric.** Finally, when a ground truth vector $\mathbf{y}$ is available, it is possible to evaluate the quality of the ranking generated by $r_{\boldsymbol{\theta}}$ on a given query-documents tuple $x \in \mathcal{Q} \times \mathcal{D}^k$. Therefore, we denote by $m : \mathbb{R}^k \times \{0,1\}^k \to \mathbb{R}$ the evaluation function that takes a vector of scores $g_{\boldsymbol{\theta}}(x)$ and a ground truth vector $\mathbf{y}$ as input and returns the corresponding metric value. Without any lack of generality, we assume that $m$ increases with the ranking quality ("higher is better").

### 2.2 Abstention in Reranking

In this work, our goal is to design an abstention mechanism built directly on top of reranker $r_{\boldsymbol{\theta}}$. We therefore aim to find a confidence function $c : \mathcal{Q} \times \mathcal{D}^k \to \mathbb{R}$ and a threshold $\tau \in \mathbb{R}$ such that ranking $r_{\boldsymbol{\theta}}(x)$ is outputted for a given query-documents tuple $x \in \mathcal{Q} \times \mathcal{D}^k$ if and only if $c(x) > \tau$. Formally, we want to come up with a function $\rho_{\boldsymbol{\theta}} : \left( \mathcal{Q} \times \mathcal{D}^k \right) \times \mathbb{R} \to \mathfrak{S}_k \cup \{\bot\}$ such that for given $x$ and $\tau$,

$$\rho_{\boldsymbol{\theta}}(x, \tau) = \begin{cases} r_{\boldsymbol{\theta}}(x), & \text{if } c(x) > \tau \\ \bot, & \text{otherwise} \end{cases} , \tag{5}$$

---

[4]Note that $\mathbf{y}$ is not a one-hot vector as several documents can be relevant to a single query.
[5]$s_j(\mathbf{z})$ denotes the $j^{\text{th}}$ element of $s(\mathbf{z})$.

where $\perp$ denotes the abstention decision.

**Requirements.** In compliance with the set of constraints defined in the introduction (Section 1), $c$ must be usable in a fully black-box setting and not incur significant extra computational costs, either during training or at inference time. Formally, we set the two following requirements:

1. We focus on the case in which the encoder parameters $\boldsymbol{\theta}$ are not learnable.
2. Furthermore, $c$ must only take relevance scores as input, i.e., $c \in \left\{ u \circ g_{\boldsymbol{\theta}} \mid u : \mathbb{R}^k \to \mathbb{R} \right\}$.

## 2.3   Related Work

Abstention, also known as selective prediction, has been around for a long time, originally for classification purposes (Chow, 1957; El-Yaniv & Wiener, 2010), and has often been associated with topics such as confidence estimation, out-of-domain (OOD) detection (Schölkopf et al., 1999; Liang et al., 2017; Darrin et al., 2022; 2024) and prediction error detection (Hendrycks & Gimpel, 2016). These fields collectively encompass three main approaches: learning-based methods, which require a dedicated training process; white-box methods, which rely on full access to the model's internal features; and black-box methods, which focus solely on the model's output.

Among learning-based methods, the most classic initiatives include algorithms such as SVMs, nearest neighbors, boosting (Hellman, 1970; Fumera & Roli, 2002; Wiener & El-Yaniv, 2015; Cortes et al., 2016) or Bayesian methods (Blundell et al., 2015) such as Markov Chain Monte Carlo (Geyer, 1992) and Variational Inference (Hinton & Van Camp, 1993; Graves, 2011), that include uncertainty estimation as a whole part of the training process. Similarly, aleatoric uncertainty estimation requires the use of a log-likelihood loss function (Loquercio et al., 2020), thus greatly modifying the training process. Some approaches also include specific regularizers in the loss function (Xin et al., 2021; Zeng et al., 2021), while others incorporate abstention as a class during training (Rajpurkar et al., 2018), most of the time with a certain cost associated (Bartlett & Wegkamp, 2008; El-Yaniv & Wiener, 2010; Cortes et al., 2016; Geifman & El-Yaniv, 2019). It comes obvious that none of these methods align with the no-training requirement set in Section 2.2.

Most white-box methods, on the other hand, do not require specific training. When considering selective prediction in neural networks, a straightforward and effective technique is to set a threshold over a confidence score derived from a pre-trained network, which was already optimized to predict all points (Cordella et al., 1995; De Stefano et al., 2000; Wiener & El-Yaniv, 2012). Monte Carlo dropout is another popular approach involving randomly deactivating neurons of a model at inference time to estimate its epistemic uncertainty (Houlsby et al., 2011; Gal et al., 2017; Smith & Gal, 2018). Others rely on a statistical analysis of the model's features (hidden layers) (Lee et al., 2018; Ren et al., 2021; Podolskiy et al., 2021; Haroush et al., 2021; Gomes et al., 2022) while ensemble-based methods (Gal & Ghahramani, 2016; Lakshminarayanan et al., 2017; Geifman & El-Yaniv, 2019) extend this idea and estimate confidence based on statistics of the ensemble model's features. The latter approaches might significantly increase inference time, which is not in line with our set of constraints (Section 2.2). Moreover, white-box access to the model cannot always be guaranteed in practical situations (e.g., API services).

Black-box approaches assume only access to the output layer of the classifier, i.e., the soft probabilities or logits. Some works estimate uncertainty by using the probability of the mode (Hendrycks & Gimpel, 2016; Lefebvre-Brossard et al., 2023; Hein et al., 2019; Hsu et al., 2020; Wang et al., 2023) while others utilize the entire logit distribution (Liu et al., 2020), the most widely known and straightforward confidence calibration technique remaining temperature scaling (Platt et al., 1999). However, it is unclear how to apply these frameworks to the ranking scenario, where relevance scores are neither soft probabilities nor logits but unscaled scalars.

Selective prediction and confidence estimation in NLP have seen targeted applications but have room for expansion across broader scenarios. Dong et al. (2018) describe a model developed to estimate confidence specifically for semantic parsing, while Kamath et al. (2020) delve into selective prediction for out-of-distribution (OOD) question answering, allowing the model to abstain from answering questions deemed too difficult or outside the training distribution. More recently, Xin et al. (2021) have applied selective prediction through loss regularization in various classification settings. Additionally, datasets such as SQuAD2.0 (Rajpurkar

et al., 2018) illustrate the challenge of unanswerable questions. In the IR field specifically, strategies like those proposed by Cheng et al. (2010) and Mao et al. (2023) allow a model to abstain at a certain cost during training by assessing the confidence level on partial ranks. Here again, such methods do not fit the black-box constraint mentioned in Section 2.2.

## 3 Confidence Assessment for Document Reranking

In this paper, as stated in Section 2.2, the confidence scorer we build only takes into account the list of relevance scores. **Our intuition is that, by analyzing the distribution of the ranker's scores, it should be possible to estimate its confidence on a given instance.** In the next two subsections, we describe two methods for calculating the confidence score: the first one is reference-free, meaning it does not require any external data, while the second one relies on access to reference data for calibration purposes.

### 3.1 Reference-Free Scenario

First, we present the reference-free approach, which can be divided into three stages (Figure 1):

1. **Observation.** We first observe the relevance scores $\mathbf{z} = g_{\boldsymbol{\theta}}(x)$ (Equation 2) for a given test instance $x = (q, d_1, \cdots, d_k) \in \mathcal{Q} \times \mathcal{D}^k$.
2. **Estimation.** We then assess confidence using a simple heuristic function $u_{\text{base}}$, based solely on these relevance scores (e.g., maximum).
3. **Thresholding.** Finally, we compare the confidence score computed in step 2 to a threshold $\tau$. If $u_{\text{base}}(\mathbf{z}) > \tau$, we make a prediction using ranker $r_{\boldsymbol{\theta}}$, otherwise, we abstain (Equation 5).

In this work, we evaluate a bunch of reference-free confidence functions, relying on simple statistics computed on the relevance scores. We focus more particularly on three functions inspired from MSP (Hendrycks & Gimpel, 2016): **(i)** the maximum relevance score $u_{\max}$, **(ii)** the standard deviation $u_{\text{std}}$, and **(iii)** the difference between the first and second highest relevance scores $u_{\text{1-2}}$ (Narayanan et al., 2012).

### 3.2 Data-Driven Scenario

In this scenario, rather than arbitrarily constructing a relevance-score-based confidence scorer, we consider that we have access to a reference set $\mathcal{S} = \left\{(x^i, \mathbf{y}^i)\right\}_{i=1}^{N}$ (Equation 1) and are therefore able to evaluate ranker $r_{\boldsymbol{\theta}}$ (Equation 4) on each of the $N$ labeled instances. We thus define dataset $\mathcal{S}_{\boldsymbol{\theta}}$:

$$\mathcal{S}_{\boldsymbol{\theta}} = \left\{(s_{\boldsymbol{\theta}}(x), m\left(g_{\boldsymbol{\theta}}(x), \mathbf{y}\right)) \mid (x, \mathbf{y}) \in \mathcal{S}\right\}. \tag{6}$$

As a reminder, $s_{\boldsymbol{\theta}}(x)$ is the ascending sort of query-document relevance scores (Equation 3) and $m\left(g_{\boldsymbol{\theta}}(x), \mathbf{y}\right)$ is the evaluation of the ranking induced by $g_{\boldsymbol{\theta}}(x)$ given ground truth $\mathbf{y}$ and metric function $m$.

An intuitive approach is to fit a simple supervised model to predict ranking performance $m\left(g_{\boldsymbol{\theta}}(x), \mathbf{y}\right)$ given a vector of sorted relevance scores $s_{\boldsymbol{\theta}}(x)$. In this section, we develop a regression-based approach (Figure 1):

0. **Calibration.** We use reference set $\mathcal{S}$ to derive $\mathcal{S}_{\boldsymbol{\theta}}$ (Equation 6) and then fit a regressor $h_{\text{reg}} : \mathbb{R}^k \to \mathbb{R}$ on $\mathcal{S}_{\boldsymbol{\theta}}$. Next, we derive $u_{\text{reg}} = h_{\text{reg}} \circ s$, the function that takes a vector of unsorted scores as input and returns the predicted ranking quality.
1. **Observation.** As in Section 3.1, we observe the relevance scores $\mathbf{z} = g_{\boldsymbol{\theta}}(x)$ for a given test instance $x$.
2. **Estimation.** We then assess confidence using the $u_{\text{reg}}$ function. Intuitively, the greater $u_{\text{reg}}(\mathbf{z})$, the more confident we are that $r_{\boldsymbol{\theta}}$ correctly ranks the documents with respect to the query.
3. **Thresholding.** As in Section 3.1, an abstention decision is finally made by comparing $u_{\text{reg}}(\mathbf{z})$ to $\tau$. In Section 5.3, we propose an approach to choose $\tau$.

In this work, we use a linear-regression-based confidence function $u_{\text{lin}}$ (Fisher, 1922). Formally, for a given unsorted vector of relevance scores $\mathbf{z}$,

$$u_{\text{lin}}(\mathbf{z}) = \beta_0 + \beta_1 s_1(\mathbf{z}) + \cdots + \beta_k s_k(\mathbf{z}),$$

where $\beta_0, \cdots, \beta_k \in \mathbb{R}$ are the coefficients fitted on $\mathcal{S}_{\boldsymbol{\theta}}$, with an $l_2$ regularization parameter $\lambda = 0.1$ (Hoerl & Kennard, 1970).[6] More data-driven confidence functions are explored in Appendix D.

## 4 Experimental Setup

In this section, we propose a novel experimental setup for assessing performance of abstention mechanisms in a black-box reranking setting. We present both our benchmark and evaluation metrics.

### 4.1 Models and Datasets

An extensive benchmark is built to evaluate the abstention mechanisms presented in Section 3. We collect six open-source reranking datasets (Lhoest et al., 2021) in three different languages, English, French and Chinese: `stackoverflowdupquestions-reranking` (Zhang et al., 2015), denoted STACKOVER-FLOW in the experiments, `askubuntudupquestions-reranking` (Lei et al., 2015), denoted ASKUBUNTU, `scidocs-reranking` (Cohan et al., 2020), denoted SCIDOCS, `mteb-fr-reranking-alloprof-s2p` (Lefebvre-Brossard et al., 2023), denoted ALLOPROF, `CMedQAv1-reranking` (Zhang et al., 2017), denoted CMEDQAV1, and `Mmarco-reranking` (Bonifacio et al., 2021), denoted MMARCO.

We also collect 22 open-source models for evaluation on each of the six datasets. These include models of various sizes, bi-encoders and cross-encoders, monolingual and multilingual: `ember-v1` (paper to be released soon), `llm-embedder` (Zhang et al., 2023), `bge-base-en-v1.5`, `bge-reranker-base/large` (Xiao et al., 2023), `multilingual-e5-small/large`, `e5-small-v2/large-v2` (Wang et al., 2022), `msmarco-MiniLM-L6-cos-v5`, `msmarco-distilbert-dot-v5`, `ms-marco-TinyBERT-L-2-v2`, `ms-marco-MiniLM-L-6-v2` (Reimers & Gurevych, 2021), stsb-TinyBERT-L-4, stsb-`distilroberta-base`, `multi-qa-distilbert-cos-v1`, `multi-qa-MiniLM-L6-cos-v1`, `all-MiniLM-L6-v2`, `all-distilroberta-v1`, `all-mpnet-base-v2`, `quora-distilroberta-base` and `qnli-distilroberta-base` (Reimers & Gurevych, 2019).[7]

In addition, each dataset is preprocessed in such a way that there are only 10 candidate documents per query and a maximum of five positive documents, in order to create a scenario close to real-life reranking use cases. Moreover, to fit the black-box setting, query-documents relevance scores are calculated upstream of the evaluation and are not used thereafter. Further details are given in appendix A.

### 4.2 Instance-Wise Metrics

In this work, we rely on three of the metrics most commonly used in the IR setting (e.g., MTEB leaderboard (Muennighoff et al., 2022)): Average Precision (AP), Normalized Discounted Cumulative Gain (NDCG) and Reciprocal Rank (RR): AP (Zhu, 2004) computes a proxy of the area under the precision-recall curve; NDCG (Järvelin & Kekäläinen, 2002) measures the relevance of the top-ranked items by discounting those further down the list using a logarithmic factor; and RR (Voorhees et al., 1999) computes the inverse rank of the first relevant element in the predicted ranking. For all three metrics, higher values indicate better performance. When relevant, we rely on their averaged versions across multiple instances: mAP, mNDCG and mRR.

### 4.3 Assessing Abstention Performance

The compromise between abstention rate and performance in predictive modeling represents a delicate balance that requires careful consideration (El-Yaniv & Wiener, 2010). First, we want to make sure that an increasing abstention rate implies increasing performance, otherwise the mechanism employed is ineffective or even deleterious. Secondly, we aim to abstain in the best possible way for every abstention rate, i.e., we want to maximize the growth of the performance-abstention curve.

Inspired by the risk-coverage curve (El-Yaniv & Wiener, 2010; Geifman & El-Yaniv, 2017), we evaluate abstention strategies by reporting the normalized Area Under the performance-abstention Curve, where the

---

[6]Scikit-learn implementation (Pedregosa et al., 2011).
[7]We take the liberty of evaluating models on languages in which they have not been trained, as we posit it is possible to detect relevant patterns in the relevance scores in any case. We discuss this assertion in Section 5.2.

x-axis corresponds to the abstention rate of the strategy and the y-axis to the achieved average performance regarding metric function $m$. Our evaluation protocol is as follows, assuming access to a test set, $\mathcal{S}'$:

1. **Multi-thresholding.** First of all, we evaluate abstention mechanism $\rho_{\boldsymbol{\theta}}$ (Equation 5) on test set $\mathcal{S}'$ for a list of abstention thresholds $\tau_1 < ... < \tau_n \in \mathbb{R}$. For a given $\tau \in \mathbb{R}$, the performance of mechanism $\rho_{\boldsymbol{\theta}}$ on $\mathcal{S}'$ and its corresponding abstention rate are respectively denoted by

$$
\begin{cases}
P_{\tau,m}\left(\rho_{\boldsymbol{\theta}}\right) = \dfrac{1}{|\mathcal{S}'_\tau|} \displaystyle\sum_{(x,\mathbf{y}) \in \mathcal{S}'_\tau} m\left(r_{\boldsymbol{\theta}}(x), \mathbf{y}\right), \\[4mm]
R_\tau\left(\rho_{\boldsymbol{\theta}}\right) = 1 - \dfrac{|\mathcal{S}'_\tau|}{|\mathcal{S}'|}
\end{cases}
, \tag{7}
$$

$\mathcal{S}'_\tau$ being the set of predicted instances, i.e., $\mathcal{S}'_\tau = \{(x,\mathbf{y}) \in \mathcal{S}' \mid \rho_{\boldsymbol{\theta}}\left(x,\tau\right) \neq \perp\}$. We collect $n$ abstention rates, $\left(R_{\tau_i}\left(\rho_{\boldsymbol{\theta}}\right)\right)_{i=1}^n$, and associated performances, $\left(P_{\tau_i,m}\left(\rho_{\boldsymbol{\theta}}\right)\right)_{i=1}^n$, and then compute the area under the performance-abstention curve $AUC_m\left(\rho_{\boldsymbol{\theta}}\right)$ for metric function $m$.

2. **Random baseline.** Second, we evaluate $\tilde{\rho}$, the mechanism that performs random (i.e., ineffective) abstention. Intuitively, random abstention shows flat performance as the abstention rate increases, equal to $P_{-\infty,m}\left(\rho_{\boldsymbol{\theta}}\right)$ (Equation 7).[8]

3. **Oracle evaluation.** Then, we evaluate the oracle $\rho^*$ on the same task. Intuitively, the oracle has access to the test labels and can therefore select the best instances for a given abstention rate. $AUC_m\left(\rho^*\right)$ thus upper-bounds $AUC_m\left(\rho_{\boldsymbol{\theta}}\right)$.

4. **Normalization.** We finally compute normalized AUC. Formally,

$$
nAUC_m\left(\rho_{\boldsymbol{\theta}}\right) = \frac{AUC_m\left(\rho_{\boldsymbol{\theta}}\right) - AUC_m\left(\tilde{\rho}\right)}{AUC_m\left(\rho^*\right) - AUC_m\left(\tilde{\rho}\right)}. \tag{8}
$$

In particular, $nAUC_m(\rho_{\boldsymbol{\theta}}) = 1$ means that abstention mechanism $\rho_{\boldsymbol{\theta}}$ reaches oracle performance. In contrast, $nAUC_m(\rho_{\boldsymbol{\theta}}) < 0$ indicates that $\rho_{\boldsymbol{\theta}}$ has a deleterious effect, with declining average ranking performance while abstention increases.

In this study, in order to guarantee consistent comparisons between abstention mechanisms, we randomly set aside 20% of the initial dataset as a test set, treating the remaining 80% as the reference set. Results, unless specified otherwise, are averaged across five random seeds.

## 5 Results

### 5.1 Abstention Performance

We report nAUC (Equation 8) for all methods described in Section 3, averaged model-wise (Table 1), and illustrate the dynamics of the performance-abstention curves for the `ember-v1` model on the STACKOVER-FLOW dataset in Figure 2.

**Abstention works.** All evaluated abstention-based methods improve downstream evaluation metrics (nAUCs greater than 0), showcasing the relevance of abstention approaches in a practical setting.

**Reference-based abstention works better.** The value of reference-based abstention approaches is demonstrated in Table 1, in which we notice that the linear-regression-based method $u_{\mathrm{lin}}$ performs on average largely better than all reference-free baselines, edging out the best baseline strategy ($u_{\mathrm{std}}$) by almost 10 points on the mAP metric.

Having shown the superiority of reference-based abstention methods across the entire benchmark, we conduct (unless otherwise specified) the following experiments on the `ember-v1` model using mAP for evaluation (main metric in the MTEB leaderboard (Muennighoff et al., 2022)), and compare $u_{\mathrm{lin}}$ to $u_{\mathrm{std}}$, respectively the best reference-based and reference-free methods mAP-wise.

Table 1: Abstention performance. nAUCs in % averaged model-wise, for each method, dataset and metric.

| Dataset | Method Metric | $u_{\max}$ | $u_{\mathrm{std}}$ | $u_{1\text{-}2}$ | $u_{\mathrm{lin}}$ |
|---|---|---|---|---|---|
| SciDocs | mAP | 49.2 | 58.7 | 4.7 | **65.4** |
| | mNDCG | 54.9 | 62.9 | 10.2 | **66.6** |
| | mRR | 80.2 | 80.3 | 41.3 | **80.5** |
| AskUbuntu | mAP | **24.1** | 16.2 | 7.3 | 22.0 |
| | mNDCG | **28.1** | 15.7 | 8.6 | 24.5 |
| | mRR | **32.7** | 16.6 | 14.0 | 26.2 |
| StackOverflow | mAP | 20.7 | 24.6 | 35.7 | **42.6** |
| | mNDCG | 21.0 | 25.0 | 35.6 | **42.6** |
| | mRR | 21.4 | 24.8 | 36.1 | **42.8** |
| Alloprof | mAP | 16.6 | 17.0 | 19.3 | **29.7** |
| | mNDCG | 16.8 | 16.9 | 18.8 | **29.3** |
| | mRR | 16.9 | 16.7 | 19.1 | **28.9** |
| CMedQAv1 | mAP | 24.4 | 23.3 | 22.3 | **30.6** |
| | mNDCG | 25.0 | 23.5 | 22.3 | **30.7** |
| | mRR | 26.5 | 24.2 | 23.4 | **31.5** |
| Mmarco | mAP | 30.1 | 31.0 | **39.4** | 34.0 |
| | mNDCG | 30.5 | 30.9 | **38.8** | 34.1 |
| | mRR | 30.1 | 31.0 | **39.6** | 34.3 |
| **Average** | mAP | 27.5 | 28.5 | 21.5 | **37.4** |
| | mNDCG | 29.4 | 29.1 | 22.4 | **38.0** |
| | mRR | 34.6 | 32.3 | 28.9 | **40.7** |

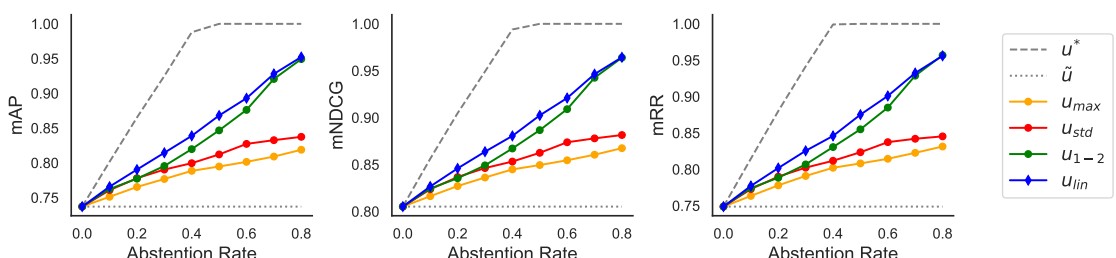

Figure 2: Performance-abstention curves for the `ember-v1` model on the StackOverflow dataset. All methods show increasing curves, indicating effective abstention. The reference-based method, $u_{\mathrm{lin}}$, stands out above the others, demonstrating superior abstention performance. Note that $u^*$ and $\tilde{u}$ respectively denote the oracle and the random baseline, associated with the $\rho^*$ and $\tilde{\rho}$ mechanisms (Sections 2.2 and 4.3).

## 5.2 Abstention Effectiveness vs. Raw Model Performance

We investigate whether a correlation exists between abstention effectiveness and model raw performance (i.e., without abstention) on the reranking task. For each model-dataset pair, we compute both no-abstention mAP and abstention nAUC, using $u_{\mathrm{lin}}$ as a confidence function (Figure 3, Table 2).

**A better ranker implies better abstention.** Figure 3 and Table 2 showcase a clear positive correlation between model raw performance and abstention effectiveness, albeit with disparities from one dataset to another. Better reranking methods naturally produce more calibrated score distributions, which in turn leads to improved abstention performance. This highlights the value of incorporating abstention mechanisms into high-performing retrieval systems and reinforces that their purpose is not to turn weak rankers into strong ones, but to enhance the quality of models that already perform reasonably well. For thoroughness, we evaluated a range of open-source models with varying performance levels, though in practice, users typically select models they already know to be sufficiently effective.

---

[8] A random (ineffective) abstention method cannot differentiate between good and bad instances. Thus, in expectation, random abstention on the test dataset results in a constant performance whatever the abstention rate.

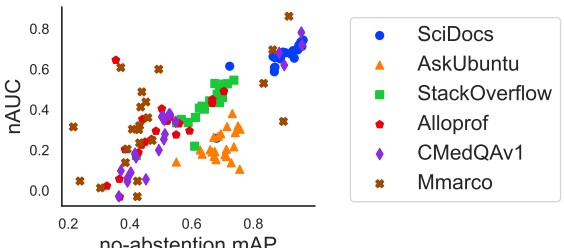

Figure 3: Abstention performance (nAUC in %) vs. no-abstention performance (mAP), for all models and datasets, using $u_{\text{lin}}$ as a confidence function. Each data point represents a model-dataset pair.

Table 2: Pearson's correlations between no-abstention mAP and abstention nAUC, for each dataset. General correlation is computed over the whole scatter plot, i.e., across all dataset-model pairs.

| Dataset | Correlation |
|---|---|
| SciDocs | 0.85 |
| AskUbuntu | 0.38 |
| StackOverflow | 0.90 |
| Alloprof | 0.56 |
| CMedQAv1 | 0.94 |
| Mmarco | 0.63 |
| **General** | **0.73** |

Table 3: MAEs for target vs. achieved abstention rates and performance levels (in %) (model: `ember-v1`, dataset: StackOverflow, metric: mAP).

| | **Target Abstention** | 10% | 50% | 90% |
|---|---|---|---|---|
| **Rate** | $u_{\text{std}}$ | 1.10 | 1.80 | 1.08 |
| | $u_{\text{lin}}$ | 1.07 | 1.79 | 1.09 |
| **mAP** | $u_{\text{std}}$ | 1.24 | 1.52 | 3.09 |
| | $u_{\text{lin}}$ | 1.22 | 1.33 | 1.51 |

Table 4: nAUCs averaged test-set-wise (model: `ember-v1`, metric: mAP).

| Method Reference set | $u_{\max}$ | $u_{\text{std}}$ | $u_{\text{1-2}}$ | $u_{\text{lin}}$ |
|---|---|---|---|---|
| Alloprof | 30.1 | **32.7** | 17.1 | 31.4 |
| AskUbuntu | 28.5 | 32.1 | 19.8 | **34.2** |
| CMedQAv1 | 27.9 | 31.1 | 17.2 | **34.2** |
| Mmarco | 27.5 | **32.2** | 18.9 | 34.9 |
| SciDocs | 23.0 | **23.5** | 22.3 | 11.6 |
| StackOverflow | 29.3 | **30.7** | 11.9 | 20.6 |
| **Average** | 27.7 | **30.4** | 17.9 | 27.8 |

## 5.3 Threshold Calibration

In practical industrial settings, a major challenge with abstention methods lies in choosing the right threshold to ensure a desired abstention rate or performance level on new data. To evaluate threshold calibration quality, we choose three target abstention rates (10%, 50%, and 90%) and rely on the reference set[9] to infer the corresponding threshold and level of performance (here, the mAP). Then, the Mean Absolute Error (MAE) between target and achieved values on test instances is reported (Table 3), averaging across 1000 random seeds on the StackOverflow dataset.[10]

$u_{\text{lin}}$ **is better calibrated than** $u_{\text{std}}$**.** The lower part of Table 3 shows that the MAE between the target and achieved mAP increases with the abstention rate, which is logical because a higher abstention rate reduces the number of evaluated instances and therefore increases volatility. However, we observe that the MAE increases much less significantly for the $u_{\text{lin}}$ method than for the $u_{\text{std}}$ method, with the latter having a MAE twice as high at a target abstention rate of 90%. This provides strong evidence that $u_{\text{lin}}$ is more reliable at high abstention rates.

## 5.4 Domain Adaptation Study

In a practical setting, it is rare to have a reference set that perfectly matches the distribution of samples seen at test time. To measure the robustness of our method to data drifts, we fit them on a given dataset and evaluate on all others.

**Reference-based approaches are globally not robust to domain changes.** From Table 4, we see on average that the reference-based method underperforms reference-free baselines when fitted on the "wrong" reference set. However, certain reference datasets seem to enable methods to generalize better overall, even

---

[9]In this setting, access to a reference set is essential in order to determine which threshold value corresponds to a given abstention rate or level of performance.

[10]For example, if the mAP corresponding to a target abstention rate of 50% in the reference set is 0.6 and the values achieved on test set on 2 different seeds are 0.5 and 0.7, the MAE is be equal to $1/2 \times (|0.6 - 0.5| + |0.6 - 0.7|) = 0.1$.

Table 5: Minimum reference set size for $u_{\text{lin}}$ to outperform $u_{\text{std}}$ (model: `ember-v1`, metric: mAP).

| Dataset | Break-Even |
|---------|------------|
| SCIDOCS | 12 |
| ASKUBUNTU | 9 |
| STACKOVERFLOW | 19 |
| ALLOPROF | 58 |
| CMEDQAv1 | 50 |
| MMARCO | 80 |
| **Average** | **38** |

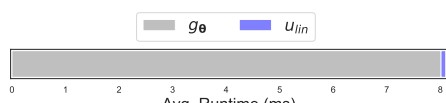

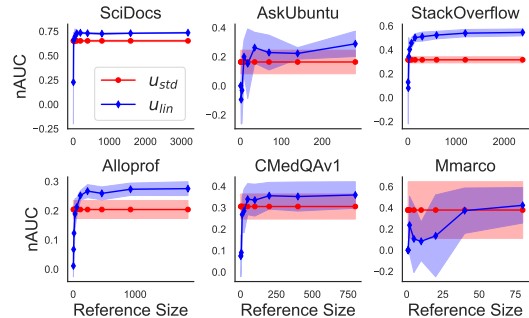

Figure 4: nAUC vs. reference set size for $u_{\text{lin}}$ and $u_{\text{std}}$ on all datasets (model: `ember-v1`, metric: mAP).

Figure 5: Comparison between relevance ($g_{\boldsymbol{\theta}}$) and confidence ($u_{\text{lin}}$) assessment times on one Apple M1 CPU (model: `all-MiniLM-L6-v2`, dataset: STACKOVERFLOW).

out of distribution, which is mainly an artifact of the number of positive documents per instance (further insights in Appendix C).

### 5.5 Reference Set Minimal Size

Calibrated reference-based methods outperform reference-free baselines (Section 5.1) but require having access to an in-domain reference set (Section 5.4). To assess the efforts required for calibrating these techniques to extract best abstention performance, we study the impact of reference set size on method performance.

**Small reference sets suffice.** From Figure 4 and Table 5, we note that $u_{\text{lin}}$ requires only a fairly small amount of reference data (less than 40 samples on average) to outperform $u_{\text{std}}$, which demonstrates the effectiveness of using a reference set even when data is scarce. This provides strong evidence on the applicability of our method to practical industrial contexts, the labeling of around 50 samples appearing perfectly tractable in such settings.

### 5.6 Computational Overhead Analysis

Crucial to the adoption of our abstention method is the minor latency overhead incurred at inference time. In our setup, we consider documents are pre-embedded by the bi-encoders, and stored in a vector database. We average timed results over 100 runs on a single instance prediction, running the calculations on one Apple M1 CPU (Figure 5). To upper-bound the abstention method overhead, we baseline the smallest model with the fastest embedding time available (`all-MiniLM-L6-v2`).

**Our data-based abstention method is free of any significant time overhead.** $u_{\text{lin}}$-based confidence estimation accounts for only 1.2% of relevance scores calculation time (Figure 5), making it an almost cost-free option.[11]

## 6 Conclusion

We developed a lightweight abstention mechanism tailored to real-world constraints, agnostic to the ranking method and shown to be effective across various settings and use cases. Our method, which only requires black-box access to relevance scores and a small reference set for calibration, is essentially a free-lunch to plug to any retrieval system, in order to gain better control over the whole IR pipeline.

---

[11]If we used a cross-encoder, documents could not be pre-embedded as they need to be appended to the input query. In that scenario, the computation time required for $g_{\boldsymbol{\theta}}$ would therefore increase, while that required for $u_{\text{lin}}$ would remain unchanged, thus reducing the relative extra running time.

## Limitations and Future Work

Our study investigates abstention mechanisms specifically within the reranking stage of an IR pipeline. While this provides a framework for estimating the confidence of the final prediction, the absence of datasets with explicit document-wise labels has limited our ability to evaluate these methods in the primary retrieval stage. These datasets typically lack labels indicating relevant and irrelevant documents, as is the case with classical retrieval benchmarks like BEIR (Thakur et al., 2021). However, in theory, our methods should be applicable at this earlier stage of the pipeline, making this a potential area for future exploration.

Moreover, our experiments have mainly focused on reranking the top-10 retrieved documents, aligning with common industrial use cases. A natural extension would be to explore how abstention mechanisms perform when applied to larger sets of documents, such as the top-50 or top-100 results. Since our linear-regression-based abstention method is designed to scale efficiently, it could potentially benefit from the richer decision-making context provided by more document scores, although this impact has yet to be quantified.

A broader consideration, however, lies in the potential of generative models, such as those used in RAG, to handle abstention on their own (Zhang et al., 2024). These models might inherently possess the capability to estimate confidence without needing a separate abstention mechanism, a compelling prospect for further investigation.

Another promising avenue is to test abstention mechanisms on relevance-scoring methods that offer more detailed information, such as token-level relevance scores for each query-document pair. Methods like ColBERT (Khattab & Zaharia, 2020) or the ColPALI (Faysse et al., 2024) model family provide richer signals, which could lead to improved abstention performance. It would be interesting to see if the additional granularity in relevance scores results in better decision-making, particularly in scenarios where token-level understanding could refine the abstention process.

## Ethical Statement

The proposed abstention mechanism enables IR systems to refrain from making predictions when signs of unreliability are detected. This approach enhances the accuracy and effectiveness of retrieval systems, but also contributes to the construction of a more trustworthy AI technology, by reducing the risks of providing potentially inaccurate or irrelevant results to users. Furthermore, abstention enables downstream systems to avoid using computational resources on unreliable or irrelevant predictions, optimizing resource utilization and therefore minimizing energy consumption. In the context of RAG systems, where large-scale computations are prevalent, introducing abstention mechanisms is a positive stride towards trustworthy and sustainable AI development, reducing overall computational demands while potentially improving system performance.

## Acknowledgements

Training compute was obtained on the Jean Zay supercomputer operated by GENCI IDRIS through compute grant 2023-AD011014668R1, AD010614770 as well as on Adastra through project c1615122, cad15031, cad14770.

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

Table 6: Details on raw datasets.

| | Reference | Language | $N$ | $k_{\min}$ | $k_{\max}$ | $\bar{k}$ | | $p_{\min}$ | $p_{\max}$ | $\bar{p}$ |
|---|---|---|---|---|---|---|---|---|---|---|
| SciDocs | Cohan et al. 2020 | English | 3978 | 26 | 60 | 29.8 | | 1 | 10 | 4.9 |
| AskUbuntu | Lei et al. 2015 | English | 375 | 20 | 20 | 20.0 | | 1 | 20 | 6.0 |
| StackOverflow | Zhang et al. 2015 | English | 2992 | 20 | 30 | 29.9 | | 1 | 6 | 1.2 |
| Alloprof | Lefebvre-Brossard et al. 2023 | French | 2316 | 10 | 35 | 10.7 | | 1 | 29 | 1.3 |
| CMedQAv1 | Zhang et al. 2017 | Chinese | 1000 | 100 | 100 | 100.0 | | 1 | 19 | 1.9 |
| Mmarco | Bonifacio et al. 2021 | Chinese | 100 | 1000 | 1002 | 1000.3 | | 1 | 3 | 1.1 |

Table 7: Dataset details after preprocessing.

| | $N$ | $k$ | $p_{\min}$ | $p_{\max}$ | $\bar{p}$ |
|---|---|---|---|---|---|
| SciDocs | 3978 | 10 | 1 | 5 | 4.9 |
| AskUbuntu | 351 | 10 | 1 | 5 | 3.6 |
| StackOverflow | 2980 | 10 | 1 | 5 | 1.2 |
| Alloprof | 2316 | 10 | 1 | 5 | 1.3 |
| CMedQAv1 | 1000 | 10 | 1 | 5 | 1.8 |
| Mmarco | 100 | 10 | 1 | 3 | 1.1 |

Table 8: Model details.

| | Reference | Languages | Type | Similarity | $|\boldsymbol{\theta}|$ |
|---|---|---|---|---|---|
| ember-v1 | TBA | English | bi | cos | 335 |
| llm-embedder | Zhang et al. 2023 | English | bi | cos | 109 |
| bge-base-en-v1.5 | Xiao et al. 2023 | English | bi | cos | 109 |
| bge-reranker-base | Xiao et al. 2023 | English, Chinese | cross | n.a. | 278 |
| bge-reranker-large | Xiao et al. 2023 | English, Chinese | cross | n.a. | 560 |
| e5-small-v2 | Wang et al. 2022 | English | bi | cos | 33 |
| e5-large-v2 | Wang et al. 2022 | English | bi | cos | 335 |
| multilingual-e5-small | Wang et al. 2022 | 94 | bi | cos | 118 |
| multilingual-e5-large | Wang et al. 2022 | 94 | bi | cos | 560 |
| msmarco-MiniLM-L6-cos-v5 | Reimers & Gurevych 2021 | English | bi | cos | 23 |
| msmarco-distilbert-dot-v5 | Reimers & Gurevych 2021 | English | bi | cos | 66 |
| ms-marco-TinyBERT-L-2-v2 | Reimers & Gurevych 2021 | English | cross | n.a. | 4 |
| ms-marco-MiniLM-L-6-v2 | Reimers & Gurevych 2021 | English | cross | n.a. | 23 |
| stsb-TinyBERT-L-4 | Reimers & Gurevych 2019 | English | cross | n.a. | 14 |
| stsb-distilroberta-base | Reimers & Gurevych 2019 | English | cross | n.a. | 82 |
| multi-qa-distilbert-cos-v1 | Reimers & Gurevych 2019 | English | bi | cos | 66 |
| multi-qa-MiniLM-L6-cos-v1 | Reimers & Gurevych 2019 | English | bi | cos | 23 |
| all-MiniLM-L6-v2 | Reimers & Gurevych 2019 | English | bi | cos | 23 |
| all-distilroberta-v1 | Reimers & Gurevych 2019 | English | bi | cos | 82 |
| all-mpnet-base-v2 | Reimers & Gurevych 2019 | English | bi | cos | 109 |
| quora-distilroberta-base | Reimers & Gurevych 2019 | English | cross | n.a. | 82 |
| qnli-distilroberta-base | Reimers & Gurevych 2019 | English | cross | n.a. | 82 |

## A    Additional Information on Models and Datasets

In this section, we give details on the datasets used in our work, before the preprocessing phase (Table 6), and after preprocessing (Table 7). As in the main text, $N$ denotes the number of instances and $k$ represents the number of candidate documents. In the same logic, $k_{\min}$, $k_{\max}$ and $\bar{k}$ respectively denote the minimum, maximum and average number of candidate documents in the dataset. In addition, $p_{\min}$, $p_{\max}$, $\bar{p}$ denote the minimum, maximum and average number of positive documents respectively. Additionally, we provide information on models' raw performance (without abstention) across the whole benchmark in Table 9.

In the data preprocessing phase, we limit the number of candidate documents to 10 and the number of positive documents per instance to five, for the sake of realism and simplicity. Intuitively, these 10 documents are to be interpreted as the output of a retriever. In practice, for each instance taken from a raw dataset, we randomly sample the positive documents (maximum of five) and complete with a random sampling of the negative documents. When a raw instance contains fewer than 10 documents, it is automatically discarded.

Additionally, we provide some details about the 22 models used in our study (Table 8): their reference paper, the language(s) in which they were trained, their type (bi-encoder or cross-encoder), the similarity function used (when applicable) and finally their number of parameters denoted $|\boldsymbol{\theta}|$ (in millions).

## B    Additional Results on Abstention Performance

In this section, we propose a comprehensive evaluation of our abstention strategies, for each model on the benchmark (Table 10). It is clear that our reference-based strategies consistently outperform all baselines, whatever the size, nature (bi- or cross-encoder) or training language of the model concerned. These results support the observations made in Section 5.1 of the main text.

Table 9: Models' raw performances (mAP, mNDCG, mRR without abstention) on the whole benchmark.

| Dataset → Metric → Model ↓ | SciDocs mAP | mNDCG | mRR | AskUbuntu mAP | mNDCG | mRR | StackOverflow mAP | mNDCG | mRR | Alloprof mAP | mNDCG | mRR | CMedQAv1 mAP | mNDCG | mRR | Mmarco mAP | mNDCG | mRR | Average mAP | mNDCG | mRR |
|---|---|---|---|---|---|---|---|---|---|---|---|---|---|---|---|---|---|---|---|---|---|
| ember-v1 | 0.96 | 0.98 | 0.99 | 0.74 | 0.85 | 0.84 | 0.74 | 0.81 | 0.75 | 0.55 | 0.66 | 0.56 | 0.51 | 0.64 | 0.55 | 0.43 | 0.57 | 0.44 | 0.65 | 0.75 | 0.69 |
| llm-embedder | 0.94 | 0.97 | 0.99 | 0.71 | 0.83 | 0.82 | 0.70 | 0.78 | 0.72 | 0.52 | 0.64 | 0.53 | 0.51 | 0.64 | 0.55 | 0.47 | 0.60 | 0.47 | 0.64 | 0.74 | 0.68 |
| bge-base-en-v1.5 | 0.93 | 0.97 | 0.98 | 0.73 | 0.84 | 0.83 | 0.72 | 0.79 | 0.73 | 0.53 | 0.65 | 0.54 | 0.53 | 0.66 | 0.57 | 0.45 | 0.58 | 0.45 | 0.65 | 0.75 | 0.68 |
| bge-reranker-base | 0.87 | 0.94 | 0.96 | 0.67 | 0.80 | 0.78 | 0.57 | 0.68 | 0.58 | 0.67 | 0.75 | 0.68 | 0.96 | 0.97 | 0.96 | 0.90 | 0.92 | 0.90 | 0.77 | 0.84 | 0.81 |
| bge-reranker-large | 0.88 | 0.95 | 0.97 | 0.71 | 0.82 | 0.81 | 0.59 | 0.69 | 0.60 | 0.70 | 0.78 | 0.72 | 0.96 | 0.97 | 0.96 | 0.92 | 0.94 | 0.92 | 0.79 | 0.86 | 0.83 |
| e5-small-v2 | 0.93 | 0.97 | 0.98 | 0.69 | 0.81 | 0.78 | 0.68 | 0.76 | 0.70 | 0.48 | 0.61 | 0.50 | 0.53 | 0.66 | 0.57 | 0.44 | 0.57 | 0.44 | 0.63 | 0.73 | 0.66 |
| e5-large-v2 | 0.95 | 0.98 | 0.99 | 0.71 | 0.83 | 0.81 | 0.69 | 0.77 | 0.70 | 0.56 | 0.67 | 0.57 | 0.55 | 0.67 | 0.59 | 0.49 | 0.61 | 0.50 | 0.66 | 0.76 | 0.69 |
| multilingual-e5-small | 0.92 | 0.96 | 0.98 | 0.69 | 0.82 | 0.80 | 0.68 | 0.76 | 0.69 | 0.59 | 0.70 | 0.61 | 0.88 | 0.92 | 0.90 | 0.83 | 0.87 | 0.83 | 0.77 | 0.84 | 0.80 |
| multilingual-e5-large | 0.94 | 0.98 | 0.99 | 0.71 | 0.83 | 0.81 | 0.69 | 0.77 | 0.70 | 0.67 | 0.75 | 0.68 | 0.90 | 0.93 | 0.92 | 0.86 | 0.90 | 0.86 | 0.80 | 0.86 | 0.83 |
| msmarco-MiniLM-L6-cos-v5 | 0.86 | 0.94 | 0.96 | 0.67 | 0.80 | 0.79 | 0.62 | 0.72 | 0.64 | 0.38 | 0.53 | 0.40 | 0.42 | 0.57 | 0.45 | 0.37 | 0.52 | 0.37 | 0.56 | 0.68 | 0.60 |
| msmarco-distilbert-dot-v5 | 0.87 | 0.94 | 0.97 | 0.68 | 0.81 | 0.79 | 0.64 | 0.73 | 0.65 | 0.44 | 0.58 | 0.45 | 0.45 | 0.60 | 0.49 | 0.43 | 0.57 | 0.44 | 0.59 | 0.70 | 0.63 |
| ms-marco-TinyBERT-L-2-v2 | 0.87 | 0.94 | 0.96 | 0.63 | 0.77 | 0.74 | 0.63 | 0.73 | 0.65 | 0.45 | 0.58 | 0.46 | 0.40 | 0.56 | 0.43 | 0.41 | 0.55 | 0.41 | 0.57 | 0.69 | 0.61 |
| ms-marco-MiniLM-L-6-v2 | 0.89 | 0.95 | 0.97 | 0.66 | 0.79 | 0.75 | 0.66 | 0.75 | 0.67 | 0.52 | 0.64 | 0.53 | 0.42 | 0.57 | 0.45 | 0.43 | 0.56 | 0.43 | 0.60 | 0.71 | 0.63 |
| stsb-TinyBERT-L-4 | 0.87 | 0.94 | 0.96 | 0.63 | 0.77 | 0.73 | 0.55 | 0.66 | 0.56 | 0.33 | 0.49 | 0.33 | 0.39 | 0.55 | 0.42 | 0.39 | 0.53 | 0.39 | 0.53 | 0.66 | 0.57 |
| stsb-distilroberta-base | 0.87 | 0.94 | 0.96 | 0.67 | 0.80 | 0.79 | 0.61 | 0.71 | 0.63 | 0.43 | 0.57 | 0.44 | 0.37 | 0.54 | 0.40 | 0.22 | 0.40 | 0.21 | 0.53 | 0.66 | 0.57 |
| multi-qa-distilbert-cos-v1 | 0.90 | 0.96 | 0.97 | 0.75 | 0.85 | 0.84 | 0.70 | 0.78 | 0.71 | 0.51 | 0.63 | 0.53 | 0.51 | 0.64 | 0.55 | 0.39 | 0.53 | 0.39 | 0.63 | 0.73 | 0.66 |
| multi-qa-MiniLM-L6-cos-v1 | 0.90 | 0.95 | 0.97 | 0.73 | 0.84 | 0.82 | 0.68 | 0.76 | 0.69 | 0.44 | 0.58 | 0.45 | 0.51 | 0.65 | 0.55 | 0.42 | 0.56 | 0.42 | 0.61 | 0.72 | 0.65 |
| all-MiniLM-L6-v2 | 0.95 | 0.98 | 0.99 | 0.74 | 0.84 | 0.83 | 0.69 | 0.77 | 0.70 | 0.35 | 0.50 | 0.36 | 0.50 | 0.64 | 0.55 | 0.46 | 0.59 | 0.46 | 0.62 | 0.72 | 0.65 |
| all-distilroberta-v1 | 0.96 | 0.98 | 0.99 | 0.76 | 0.85 | 0.85 | 0.70 | 0.78 | 0.71 | 0.47 | 0.60 | 0.49 | 0.49 | 0.63 | 0.54 | 0.43 | 0.57 | 0.44 | 0.63 | 0.74 | 0.67 |
| all-mpnet-base-v2 | 0.96 | 0.98 | 0.99 | 0.76 | 0.85 | 0.85 | 0.70 | 0.78 | 0.71 | 0.50 | 0.63 | 0.52 | 0.51 | 0.65 | 0.55 | 0.42 | 0.56 | 0.42 | 0.64 | 0.74 | 0.67 |
| quora-distilroberta-base | 0.72 | 0.86 | 0.89 | 0.67 | 0.80 | 0.79 | 0.61 | 0.71 | 0.62 | 0.37 | 0.52 | 0.38 | 0.39 | 0.55 | 0.43 | 0.24 | 0.41 | 0.24 | 0.50 | 0.64 | 0.56 |
| qnli-distilroberta-base | 0.68 | 0.82 | 0.80 | 0.55 | 0.71 | 0.63 | 0.42 | 0.56 | 0.43 | 0.37 | 0.52 | 0.38 | 0.36 | 0.53 | 0.40 | 0.30 | 0.46 | 0.30 | 0.45 | 0.60 | 0.49 |

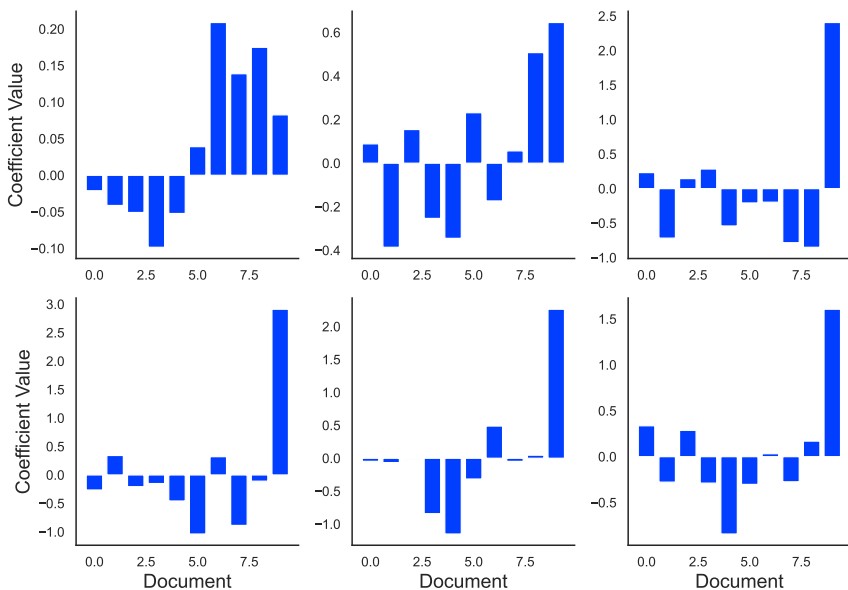

Figure 6: Coefficients of the $u_{\text{lin}}$ confidence function (model: `ember-v1`, metric: mAP). Recall that documents are sorted in increasing order of relevance scores. Therefore, document 10 corresponds to the document with highest relevance score while document 1 is the least relevant to the query.

## C   Additional Results on Domain Adaptation

In this section, we provide additional results concerning domain adaptation. In particular, we propose to take a closer look at the coefficients of our $u_{\text{lin}}$ confidence function (Figure 6) and to relate this information to abstention performance on the whole benchmark (Table 11).

A simple observation that can be made from Figure 6 is that the value taken by the coefficients depends very strongly on the number of positive documents per instance (see Table 7). Indeed, when we look at the results for the SciDocs dataset (five positive documents per instance on average), we see that $u_{\text{lin}}$ has positive weights on the first five documents and negative weights on the last five. A similar observation can be made for the other five datasets.

These observations are corroborated by Table 11, which shows that domain adaptation works well when the average number of positive documents per instance is approximately the same (e.g., from StackOverflow to Alloprof). On the contrary, we can see that when calibration is performed on SciDocs, performance transfers much less well, since the rest of the datasets have a significantly lower number of positive documents

Table 10: Methods' nAUCs (in %) on the whole benchmark.

| Model ↓ / Metric ↓ | SciDocs $u_{max}$ | $u_{std}$ | $u_{1-2}$ | $u_{lin}$ | AskUbuntu $u_{max}$ | $u_{std}$ | $u_{1-2}$ | $u_{lin}$ | StackOverflow $u_{max}$ | $u_{std}$ | $u_{1-2}$ | $u_{lin}$ | Alloprof $u_{max}$ | $u_{std}$ | $u_{1-2}$ | $u_{lin}$ | CMedQAv1 $u_{max}$ | $u_{std}$ | $u_{1-2}$ | $u_{lin}$ | Mmarco $u_{max}$ | $u_{std}$ | $u_{1-2}$ | $u_{lin}$ |
|---|---|---|---|---|---|---|---|---|---|---|---|---|---|---|---|---|---|---|---|---|---|---|---|---|
| ember-v1 mAP | 50.7 | 65.2 | -4.7 | 73.6 | 30.2 | 16.3 | 10.2 | 28.6 | 23.7 | 31.7 | 48.0 | 54.6 | 16.4 | 20.4 | 20.8 | 27.6 | 28.2 | 30.5 | 23.6 | 35.9 | 42.2 | 36.4 | 37.9 | 41.3 |
| mNDCG | 56.8 | 68.4 | 0.4 | 73.8 | 35.2 | 13.9 | 14.2 | 30.8 | 24.3 | 32.3 | 47.8 | 54.0 | 16.9 | 20.4 | 20.2 | 27.3 | 27.5 | 29.7 | 22.3 | 34.3 | 41.4 | 35.8 | 36.8 | 40.4 |
| mRR | 83.4 | 81.1 | 34.6 | 83.6 | 41.1 | 11.3 | 22.2 | 31.0 | 25.0 | 32.0 | 48.1 | 54.0 | 17.0 | 19.8 | 20.8 | 26.9 | 27.2 | 30.6 | 21.7 | 33.6 | 42.1 | 36.8 | 36.8 | 41.5 |
| llm-embedder mAP | 47.2 | 62.3 | 1.9 | 69.9 | 31.7 | 20.7 | 1.7 | 31.6 | 26.0 | 32.3 | 47.1 | 52.6 | 19.7 | 22.7 | 22.8 | 34.5 | 22.4 | 29.8 | 21.5 | 36.7 | 18.0 | 25.5 | 40.5 | 24.7 |
| mNDCG | 52.9 | 66.4 | 5.8 | 69.9 | 36.7 | 19.4 | 2.1 | 34.7 | 26.5 | 32.8 | 47.0 | 52.7 | 20.0 | 22.6 | 22.5 | 34.4 | 23.0 | 29.9 | 20.6 | 36.0 | 18.8 | 25.4 | 40.6 | 25.5 |
| mRR | 83.1 | 85.4 | 36.7 | 86.4 | 42.0 | 19.8 | 6.4 | 40.3 | 26.8 | 32.5 | 47.8 | 53.2 | 19.7 | 22.5 | 23.0 | 34.5 | 25.1 | 31.4 | 21.6 | 37.5 | 19.0 | 26.4 | 39.8 | 25.7 |
| bge-base-en-v1.5 mAP | 46.4 | 58.2 | -1.4 | 65.4 | 35.2 | 14.2 | 0.7 | 38.1 | 23.9 | 32.3 | 46.4 | 52.8 | 17.1 | 25.0 | 27.8 | 36.4 | 29.1 | 34.4 | 24.4 | 37.7 | 40.0 | 27.7 | 38.2 | 43.8 |
| mNDCG | 53.3 | 62.1 | 4.7 | 66.5 | 39.2 | 10.2 | 0.5 | 38.8 | 24.5 | 32.8 | 46.1 | 53.1 | 17.2 | 24.8 | 26.8 | 35.3 | 28.8 | 34.2 | 23.2 | 36.6 | 40.5 | 27.0 | 36.2 | 45.2 |
| mRR | 83.7 | 79.8 | 40.4 | 83.1 | 41.7 | 3.1 | 3.8 | 37.6 | 24.8 | 32.5 | 46.4 | 52.6 | 16.6 | 24.4 | 26.6 | 34.9 | 28.6 | 34.1 | 23.8 | 36.2 | 39.6 | 26.6 | 38.5 | 43.3 |
| bge-reranker-base mAP | 47.1 | 53.8 | 2.7 | 58.7 | 24.2 | 8.1 | -8.9 | 19.0 | 20.7 | 24.1 | 26.7 | 35.2 | 25.1 | 30.4 | 38.2 | 45.0 | 74.9 | 61.4 | 65.1 | 78.2 | 51.3 | 38.1 | 65.8 | 34.2 |
| mNDCG | 51.3 | 57.0 | 8.5 | 59.2 | 29.4 | 7.7 | -6.7 | 21.1 | 20.7 | 24.4 | 26.6 | 34.9 | 25.4 | 30.8 | 37.7 | 44.3 | 78.1 | 64.3 | 67.3 | 79.7 | 53.1 | 40.5 | 64.4 | 37.3 |
| mRR | 67.0 | 68.1 | 36.4 | 66.9 | 37.5 | 13.0 | 0.4 | 22.9 | 20.8 | 24.3 | 27.5 | 35.9 | 25.3 | 30.7 | 38.0 | 44.0 | 82.3 | 67.5 | 70.9 | 83.6 | 53.5 | 42.3 | 64.6 | 35.3 |
| bge-reranker-large mAP | 54.2 | 59.6 | 12.0 | 65.5 | 23.7 | 24.4 | -1.7 | 17.7 | 17.1 | 24.0 | 24.4 | 33.6 | 25.5 | 32.6 | 44.0 | 49.0 | 70.9 | 56.9 | 59.6 | 71.9 | 75.8 | 75.1 | 85.2 | 86.2 |
| mNDCG | 59.3 | 64.0 | 18.0 | 66.3 | 25.3 | 23.9 | -1.5 | 15.5 | 17.0 | 24.1 | 24.2 | 33.6 | 25.9 | 32.9 | 43.6 | 48.8 | 74.5 | 59.6 | 61.3 | 75.8 | 76.3 | 75.5 | 85.3 | 86.4 |
| mRR | 80.4 | 81.4 | 49.1 | 79.9 | 26.5 | 24.8 | 4.9 | 6.1 | 17.3 | 24.2 | 24.6 | 33.9 | 26.0 | 32.5 | 44.0 | 48.8 | 81.0 | 64.6 | 64.6 | 78.5 | 75.8 | 75.1 | 85.2 | 86.2 |
| e5-small-v2 mAP | 49.1 | 61.8 | 1.1 | 69.2 | 23.1 | 15.8 | 16.6 | 26.9 | 22.4 | 27.4 | 38.3 | 47.5 | 12.2 | 17.9 | 20.5 | 29.4 | 30.8 | 32.9 | 19.9 | 37.5 | 46.7 | 47.3 | 50.2 | 48.8 |
| mNDCG | 56.1 | 66.9 | 7.3 | 70.7 | 26.8 | 14.3 | 18.7 | 29.2 | 22.8 | 28.0 | 38.3 | 47.5 | 12.7 | 17.9 | 20.0 | 29.0 | 30.5 | 32.9 | 20.3 | 37.4 | 46.3 | 46.7 | 49.1 | 46.3 |
| mRR | 86.6 | 88.3 | 44.2 | 87.3 | 27.6 | 10.6 | 26.5 | 29.4 | 23.4 | 27.7 | 38.8 | 48.1 | 13.5 | 17.4 | 19.8 | 28.7 | 30.8 | 32.9 | 22.0 | 37.4 | 46.4 | 46.9 | 50.9 | 48.5 |
| e5-large-v2 mAP | 52.0 | 63.7 | 3.2 | 71.9 | 17.6 | 16.2 | 12.9 | 20.4 | 23.3 | 27.6 | 39.2 | 47.2 | 18.3 | 23.9 | 20.6 | 33.1 | 26.8 | 30.2 | 22.7 | 34.0 | 52.6 | 54.0 | 64.2 | 60.0 |
| mNDCG | 58.1 | 68.0 | 8.0 | 72.9 | 22.9 | 15.7 | 16.5 | 23.9 | 23.6 | 28.1 | 39.1 | 47.5 | 18.4 | 23.5 | 19.8 | 32.6 | 26.3 | 29.7 | 22.6 | 33.4 | 53.0 | 54.2 | 64.0 | 60.2 |
| mRR | 87.6 | 88.2 | 41.6 | 85.8 | 27.1 | 14.1 | 27.1 | 24.4 | 23.9 | 28.1 | 40.1 | 48.1 | 18.7 | 22.9 | 19.8 | 32.3 | 24.7 | 28.2 | 23.9 | 32.2 | 52.4 | 53.7 | 64.4 | 59.4 |
| multilingual-e5-small mAP | 46.1 | 59.3 | 4.1 | 67.6 | 29.8 | 20.3 | -0.6 | 28.0 | 25.4 | 31.4 | 45.5 | 52.9 | 8.5 | 22.3 | 20.4 | 29.5 | 61.1 | 56.1 | 48.3 | 68.1 | 62.8 | 43.5 | 86.0 | 53.0 |
| mNDCG | 52.2 | 63.7 | 9.1 | 68.6 | 34.1 | 22.9 | 1.2 | 33.7 | 25.7 | 31.8 | 45.4 | 53.0 | 8.7 | 22.2 | 19.8 | 28.7 | 63.7 | 57.5 | 50.1 | 70.1 | 63.5 | 44.6 | 86.0 | 52.8 |
| mRR | 82.7 | 86.4 | 41.8 | 84.9 | 37.6 | 29.7 | 5.5 | 37.0 | 25.5 | 31.4 | 45.8 | 53.6 | 9.3 | 21.1 | 19.7 | 27.2 | 67.8 | 59.5 | 54.0 | 73.9 | 62.9 | 43.3 | 85.9 | 53.0 |
| multilingual-e5-large mAP | 49.4 | 62.6 | 2.8 | 69.5 | 17.2 | 10.5 | -16.2 | 16.8 | 23.0 | 27.5 | 43.2 | 50.2 | 16.9 | 27.7 | 37.5 | 43.1 | 59.3 | 47.3 | 48.9 | 62.0 | 74.3 | 52.7 | 87.2 | 69.6 |
| mNDCG | 55.8 | 66.9 | 7.8 | 70.9 | 21.5 | 10.7 | -16.3 | 18.7 | 23.4 | 28.0 | 43.0 | 50.3 | 17.4 | 27.8 | 37.1 | 42.7 | 63.5 | 49.4 | 51.1 | 65.3 | 75.2 | 54.3 | 86.8 | 68.8 |
| mRR | 88.0 | 86.7 | 40.7 | 86.2 | 24.9 | 13.2 | -13.8 | 17.7 | 23.7 | 28.1 | 43.7 | 50.8 | 18.1 | 27.1 | 37.1 | 42.7 | 70.2 | 52.0 | 56.7 | 70.0 | 76.5 | 57.1 | 87.3 | 69.4 |
| msmarco-MiniLM-L6-cos-v5 mAP | 51.5 | 59.9 | 1.4 | 66.3 | 19.5 | 12.0 | 10.1 | 19.3 | 22.1 | 22.1 | 36.0 | 40.7 | 10.6 | 10.0 | 15.1 | 20.3 | 11.0 | 9.9 | 16.9 | 18.2 | 21.9 | 26.8 | 49.3 | 60.8 |
| mNDCG | 57.0 | 63.1 | 7.6 | 67.4 | 24.3 | 10.1 | 10.3 | 22.4 | 22.6 | 22.7 | 36.0 | 41.0 | 10.8 | 9.8 | 14.1 | 19.3 | 10.5 | 9.0 | 16.1 | 17.6 | 21.9 | 26.3 | 48.2 | 58.1 |
| mRR | 79.5 | 74.9 | 40.4 | 79.7 | 28.4 | 6.6 | 14.9 | 24.3 | 23.4 | 23.3 | 36.6 | 41.8 | 11.1 | 9.8 | 14.0 | 17.8 | 10.2 | 7.8 | 17.0 | 18.5 | 22.1 | 25.4 | 48.7 | 59.6 |
| msmarco-distilbert-dot-v5 mAP | 50.3 | 59.7 | 2.3 | 66.1 | 15.1 | 13.0 | 2.1 | 15.2 | 20.8 | 24.3 | 36.4 | 40.2 | 11.4 | 16.2 | 8.7 | 22.5 | 1.3 | 8.4 | 6.8 | 5.5 | -0.1 | 21.8 | 54.6 | 32.1 |
| mNDCG | 55.5 | 63.3 | 8.0 | 67.4 | 20.1 | 13.1 | 3.0 | 17.9 | 21.1 | 24.7 | 36.3 | 40.1 | 11.8 | 15.6 | 8.4 | 22.2 | 2.1 | 8.2 | 6.4 | 5.2 | 1.0 | 20.6 | 53.6 | 29.8 |
| mRR | 77.7 | 76.4 | 41.0 | 79.5 | 26.6 | 13.7 | 11.1 | 24.9 | 21.2 | 24.6 | 37.2 | 40.9 | 11.4 | 15.9 | 8.8 | 22.1 | 3.6 | 6.1 | 7.5 | 6.2 | 0.2 | 19.9 | 53.4 | 28.8 |
| ms-marco-TinyBERT-L-2-v2 mAP | 53.6 | 55.6 | 27.6 | 60.7 | 26.5 | 27.3 | 0.6 | 18.0 | 24.4 | 20.0 | 32.2 | 41.7 | 12.1 | 7.9 | 14.0 | 24.0 | 5.4 | -9.1 | -2.3 | 8.9 | 26.5 | 45.0 | 29.0 | 30.3 |
| mNDCG | 59.2 | 61.0 | 35.4 | 62.9 | 29.4 | 29.0 | -0.3 | 22.4 | 24.5 | 20.5 | 31.9 | 41.5 | 12.4 | 7.9 | 13.6 | 23.8 | 4.7 | -8.9 | -3.3 | 8.0 | 26.8 | 43.5 | 27.4 | 29.5 |
| mRR | 85.8 | 85.4 | 76.1 | 83.4 | 30.3 | 28.4 | 2.4 | 21.9 | 24.5 | 20.2 | 32.4 | 41.7 | 12.2 | 8.0 | 14.1 | 23.8 | 3.4 | -7.5 | -2.9 | 5.5 | 27.2 | 44.2 | 29.8 | 31.2 |
| ms-marco-MiniLM-L-6-v2 mAP | 52.6 | 56.3 | 16.1 | 65.1 | 21.7 | 24.3 | 5.0 | 20.1 | 18.8 | 21.3 | 38.3 | 44.3 | 24.7 | 29.3 | 19.9 | 34.4 | 9.2 | 4.1 | 13.2 | 16.4 | -6.8 | 32.9 | 18.7 | 30.1 |
| mNDCG | 58.8 | 61.9 | 25.1 | 66.6 | 25.9 | 27.3 | 3.6 | 23.1 | 19.2 | 21.9 | 38.0 | 44.3 | 24.7 | 29.0 | 19.4 | 34.1 | 9.2 | 3.5 | 13.4 | 16.3 | -4.9 | 32.4 | 17.3 | 30.8 |
| mRR | 87.4 | 86.9 | 72.4 | 86.4 | 29.5 | 32.7 | 6.2 | 20.6 | 19.4 | 21.9 | 38.6 | 44.6 | 25.0 | 29.1 | 20.2 | 33.9 | 8.1 | 4.5 | 14.5 | 16.6 | -7.3 | 32.4 | 19.7 | 28.1 |
| stsb-TinyBERT-L-4 mAP | 45.2 | 56.4 | -8.8 | 61.0 | 14.7 | 13.6 | 7.3 | 20.0 | 16.3 | 15.9 | 25.7 | 33.5 | 7.2 | -1.8 | -1.1 | 2.3 | -2.0 | 3.3 | 1.5 | 6.1 | 13.0 | 22.5 | 21.0 | 13.8 |
| mNDCG | 51.2 | 60.5 | -4.0 | 62.2 | 16.4 | 12.2 | 9.3 | 18.6 | 16.6 | 16.3 | 25.3 | 33.3 | 7.7 | -2.3 | -1.9 | 2.5 | -1.4 | 4.0 | 1.5 | 6.0 | 12.2 | 21.1 | 20.0 | 17.1 |
| mRR | 73.2 | 73.6 | 20.7 | 73.5 | 21.4 | 12.9 | 17.6 | 18.3 | 17.7 | 16.6 | 25.7 | 33.5 | 8.0 | -2.6 | -1.8 | 3.4 | -1.3 | 5.4 | 0.2 | 7.9 | 12.0 | 21.2 | 20.1 | 17.1 |
| stsb-distilroberta-base mAP | 49.2 | 56.8 | 3.8 | 65.7 | 19.3 | 7.6 | 6.5 | 16.9 | 21.8 | 23.1 | 28.1 | 36.2 | -4.7 | 11.2 | 4.7 | 18.7 | 15.0 | -0.5 | 11.1 | 9.7 | 11.4 | -0.8 | -1.1 | 31.5 |
| mNDCG | 55.0 | 61.8 | 11.4 | 66.9 | 21.1 | 7.3 | 8.8 | 20.2 | 21.8 | 23.5 | 27.9 | 36.6 | -4.9 | 10.5 | 4.5 | 18.9 | 14.1 | -1.8 | 10.3 | 9.6 | 11.2 | -0.9 | 0.2 | 32.7 |
| mRR | 78.6 | 80.4 | 50.4 | 80.7 | 23.9 | 12.2 | 14.2 | 20.7 | 22.1 | 23.5 | 28.6 | 36.9 | -5.7 | 9.1 | 4.1 | 18.0 | 15.6 | -1.8 | 10.8 | 8.7 | 8.8 | -2.3 | -0.4 | 31.0 |
| multi-qa-distilbert-cos-v1 mAP | 52.3 | 60.5 | -1.4 | 68.1 | 32.9 | 27.7 | 20.7 | 31.9 | 19.4 | 24.1 | 38.6 | 45.9 | 27.0 | 24.6 | 21.2 | 34.6 | 12.1 | 18.3 | 20.4 | 25.3 | 29.0 | 30.9 | 40.5 | 20.5 |
| mNDCG | 58.6 | 65.2 | 4.3 | 68.8 | 38.0 | 25.8 | 22.6 | 32.5 | 19.6 | 24.7 | 38.6 | 46.0 | 27.5 | 24.7 | 20.7 | 34.4 | 12.7 | 17.9 | 19.8 | 25.8 | 28.8 | 30.7 | 39.4 | 21.8 |
| mRR | 83.7 | 82.5 | 35.9 | 81.9 | 46.5 | 26.0 | 27.1 | 35.0 | 20.1 | 24.5 | 39.4 | 46.3 | 27.6 | 25.1 | 20.4 | 34.0 | 14.6 | 18.0 | 21.5 | 27.8 | 29.0 | 30.5 | 40.6 | 20.0 |
| multi-qa-MiniLM-L6-cos-v1 mAP | 52.7 | 60.6 | -2.1 | 68.5 | 25.1 | 6.7 | 13.0 | 13.7 | 19.7 | 27.1 | 42.0 | 47.5 | 32.4 | 23.2 | 19.0 | 35.3 | 26.6 | 30.4 | 19.7 | 34.9 | 15.3 | 21.6 | 22.1 | -3.0 |
| mNDCG | 59.6 | 66.1 | 4.6 | 70.5 | 30.7 | 3.0 | 14.6 | 16.1 | 20.2 | 27.1 | 41.8 | 47.5 | 32.3 | 22.8 | 18.4 | 34.9 | 27.1 | 30.4 | 18.5 | 34.7 | 15.3 | 21.1 | 22.2 | -4.4 |
| mRR | 88.2 | 87.6 | 40.4 | 86.9 | 40.2 | 1.6 | 19.2 | 23.4 | 20.9 | 27.1 | 42.4 | 47.7 | 33.0 | 23.3 | 18.8 | 35.2 | 28.5 | 31.5 | 18.8 | 35.4 | 15.5 | 22.4 | 22.2 | -0.2 |
| all-MiniLM-L6-v2 mAP | 52.5 | 66.5 | -6.3 | 73.4 | 23.6 | 13.5 | 6.0 | 21.6 | 19.0 | 25.1 | 35.6 | 43.5 | 37.1 | -8.3 | 28.3 | 64.5 | 28.6 | 25.0 | 30.5 | 36.5 | 23.2 | 11.2 | 27.6 | 36.0 |
| mNDCG | 59.7 | 71.8 | -2.2 | 74.9 | 29.4 | 13.5 | 6.7 | 24.3 | 19.3 | 25.7 | 35.5 | 43.4 | 37.3 | -7.0 | 27.2 | 63.1 | 28.6 | 24.1 | 29.6 | 36.6 | 24.5 | 11.3 | 27.6 | 36.1 |
| mRR | 89.8 | 93.3 | 24.0 | 91.3 | 38.8 | 21.6 | 13.9 | 32.5 | 20.3 | 25.6 | 36.5 | 43.6 | 37.9 | -7.4 | 28.4 | 61.9 | 28.8 | 24.1 | 27.9 | 36.0 | 23.3 | 10.3 | 29.6 | 37.8 |
| all-distilroberta-v1 mAP | 49.9 | 63.3 | -5.8 | 71.3 | 26.1 | 9.0 | 25.2 | 10.5 | 16.5 | 24.4 | 41.7 | 47.1 | 20.1 | 17.6 | 17.2 | 25.3 | 3.5 | 12.4 | 15.5 | 20.0 | 19.7 | 18.0 | 25.9 | 23.5 |
| mNDCG | 56.5 | 67.7 | -0.5 | 72.0 | 31.9 | 8.3 | 26.5 | 16.3 | 16.7 | 25.0 | 41.6 | 47.3 | 19.9 | 17.2 | 16.8 | 25.0 | 4.9 | 12.4 | 15.4 | 18.8 | 19.0 | 17.3 | 24.9 | 21.0 |
| mRR | 93.7 | 89.1 | 36.8 | 91.1 | 42.2 | 12.2 | 32.9 | 24.1 | 17.1 | 24.7 | 42.1 | 47.4 | 19.2 | 16.9 | 17.0 | 24.1 | 7.9 | 13.0 | 15.5 | 18.5 | 19.0 | 18.3 | 26.7 | 25.2 |
| all-mpnet-base-v2 mAP | 51.0 | 64.9 | -11.4 | 74.4 | 34.6 | 19.7 | 19.2 | 30.3 | 20.0 | 24.1 | 43.4 | 50.6 | 27.2 | 22.2 | 28.4 | 40.4 | 19.3 | 22.2 | 19.5 | 27.8 | 7.6 | 18.8 | 18.9 | 4.7 |
| mNDCG | 57.0 | 69.7 | -9.8 | 75.7 | 39.1 | 17.9 | 22.5 | 33.7 | 20.7 | 24.8 | 43.4 | 50.4 | 27.6 | 22.2 | 27.7 | 40.0 | 18.7 | 21.4 | 18.5 | 25.7 | 8.6 | 19.8 | 18.7 | 6.3 |
| mRR | 89.0 | 93.6 | 10.9 | 90.2 | 45.1 | 19.1 | 31.5 | 37.0 | 21.6 | 24.5 | 44.0 | 50.6 | 27.8 | 22.0 | 28.3 | 40.4 | 19.1 | 21.3 | 18.4 | 24.6 | 8.2 | 19.5 | 20.2 | 7.7 |
| quora-distilroberta-base mAP | 60.7 | 61.0 | 56.3 | 61.5 | 27.6 | 27.6 | 25.8 | 26.3 | 18.0 | 17.7 | 20.5 | 21.9 | -1.3 | -0.9 | -1.9 | 5.7 | 7.9 | 6.8 | 6.3 | 4.4 | 20.2 | 25.6 | 3.8 | 4.7 |
| mNDCG | 64.5 | 64.8 | 60.9 | 65.0 | 31.1 | 31.2 | 29.2 | 30.6 | 17.9 | 17.7 | 20.4 | 21.9 | -1.6 | -1.5 | -1.4 | 4.9 | 8.4 | 6.8 | 6.0 | 5.4 | 18.9 | 24.1 | 2.8 | 5.7 |
| mRR | 76.6 | 76.5 | 75.8 | 76.7 | 32.6 | 32.8 | 31.1 | 32.5 | 17.6 | 17.3 | 20.0 | 21.2 | -1.2 | -1.0 |  | 3.6 | 9.8 | 7.2 | 6.5 | 6.9 | 19.4 | 24.5 | 4.4 | 4.6 |
| qnli-distilroberta-base mAP | 19.3 | 23.8 | 10.7 | 25.8 | 11.2 | 8.5 | 3.9 | 14.1 | 13.8 | 12.7 | 8.9 | 17.3 | 2.2 | 0.5 | -1.1 | -3.2 | -5.5 | 2.3 | -1.7 | -2.9 | 17.6 | 8.3 | 1.2 | 1.3 |
| mNDCG | 19.0 | 23.0 | 13.2 | 25.6 | 10.3 | 6.8 | 4.6 | 14.7 | 13.6 | 12.5 | 8.1 | 16.9 | 2.1 | 0.8 | -1.0 | -2.2 | -4.8 | 2.0 | -1.5 | -3.6 | 18.1 | 7.9 | 1.7 | 2.6 |
| mRR | 18.6 | 21.2 | 18.8 | 24.9 | 8.9 | 6.8 | 2.8 | 15.1 | 13.6 | 12.0 | 8.0 | 16.0 | 1.5 | 0.6 | -0.7 | -1.9 | -3.4 | 1.3 | 0.2 | -3.7 | 16.6 | 8.0 | 2.2 | 2.3 |

per instance. We therefore stress again the importance of using a reference set drawn from the same distribution as the test instances to avoid this kind of pitfall, since we remind that the number of positive documents per instance is unknown to the user.

# D   Additional Reference-Based Confidence Functions

In this section, we introduce other functions and approaches to tackle confidence assessment in the reference-based setting.

## D.1   Additional Regression-Based Confidence Functions

We propose two additional regression-based confidence functions (Section 3.2): $u_{rf}$ and $u_{mlp}$. $u_{rf}$ is based on a random forest (Ho, 1995) fitted using 100 independent estimators and squared error impurity criterion. 100 was chosen among several values (10, 50, 100, 200, 500, 1000) as the best compromise between efficiency and effectiveness regarding nAUC. $u_{mlp}$ is based on a Multi-Layer Perceptron (MLP) (Rumelhart et al.,

Table 11: nAUCs (in %) for all reference-test relevant pairs (model: `ember-v1`, metric: mAP).

| Reference Set | Test Set | Method | $u_{\max}$ | $u_{\text{std}}$ | $u_{\text{1-2}}$ | $u_{\text{lin}}$ |
|---|---|---|---|---|---|---|
| SciDocs | AskUbuntu | | 23.8 | 21.7 | 8.1 | 24.6 |
| | StackOverflow | | 19.6 | 28.6 | 47.6 | -10.0 |
| | Alloprof | | 15.9 | 18.8 | 21.5 | 8.6 |
| | CedQAv1 | | 27.0 | 26.9 | 21.3 | 17.6 |
| | Mmarco | | 28.8 | 21.3 | 13.0 | 17.3 |
| AskUbuntu | SciDocs | | 51.2 | 64.9 | -4.3 | 62.8 |
| | StackOverflow | | 19.6 | 28.6 | 47.6 | 26.5 |
| | Alloprof | | 15.9 | 18.8 | 21.5 | 21.8 |
| | CMedQAv1 | | 27.0 | 26.9 | 21.3 | 30.1 |
| | Mmarco | | 28.8 | 21.3 | 13.0 | 29.9 |
| StackOverflow | SciDocs | | 51.2 | 64.9 | -4.3 | 15.5 |
| | AskUbuntu | | 23.8 | 21.7 | 8.1 | 15.1 |
| | Alloprof | | 15.9 | 18.8 | 21.5 | 25.7 |
| | CMedQAv1 | | 27.0 | 26.9 | 21.3 | 28.8 |
| | Mmarco | | 28.8 | 21.3 | 13.0 | 18.0 |
| Alloprof | SciDocs | | 51.2 | 64.9 | -4.3 | 34.1 |
| | AskUbuntu | | 23.8 | 21.7 | 8.1 | 20.9 |
| | StackOverflow | | 19.6 | 28.6 | 47.6 | 48.1 |
| | CMedQAv1 | | 27.0 | 26.9 | 21.3 | 32.7 |
| | Mmarco | | 28.8 | 21.3 | 13.0 | 21.4 |
| CMedQAv1 | SciDocs | | 51.2 | 64.9 | -4.3 | 55.9 |
| | AskUbuntu | | 23.8 | 21.7 | 8.1 | 23.0 |
| | StackOverflow | | 19.6 | 28.6 | 47.6 | 43.8 |
| | Alloprof | | 15.9 | 18.8 | 21.5 | 27.5 |
| | Mmarco | | 28.8 | 21.3 | 13.0 | 20.7 |
| Mmarco | SciDocs | | 51.2 | 64.9 | -4.3 | 44.9 |
| | AskUbuntu | | 23.8 | 21.7 | 8.1 | 23.2 |
| | StackOverflow | | 19.6 | 28.6 | 47.6 | 44.7 |
| | Alloprof | | 15.9 | 18.8 | 21.5 | 28.1 |
| | CMedQAv1 | | 27.0 | 26.9 | 21.3 | 33.8 |

1986) with one hidden layer of size 128 (retained among several values: 32, 64, 128, 256), ReLU activation, and mean squared error loss function. 0.05 was chosen among multiple learning rate values (0.001, 0.005, 0.01, 0.05, 0.1), a batch size equal to that of the reference set was selected (one single iteration per epoch), and 500 training iterations were performed, as it showed the best efficiency-effectiveness trade-off in terms of downstream nAUC.

Results using these two functions are reported in Table 12. We observe that although these two new methods logically outperform the reference-free baselines, they are still less effective than simple linear regression. This is not entirely surprising, given the limited data and constrained feature space at stake in this setup.

## D.2 Other Approaches

### D.2.1 Classification-Based Approach

Confidence assessment can be seen as a classification problem in which the good instances constitute the positive class, the bad instances constitute the negative class and the rest belongs to a neutral class. We therefore need to slightly modify reference set $\mathcal{S}_{\boldsymbol{\theta}}$ (Equation 6) to make it suitable for the classification setting. We thus build

$$\mathcal{S}_{\boldsymbol{\theta}}^{\text{clf}} = \big\{ \big( \mathbf{z}, \mathbb{1}_{\{y>m^+\}} - \mathbb{1}_{\{y<m^-\}} \big) \mid (\mathbf{z}, y) \in \mathcal{S}_{\boldsymbol{\theta}} \big\}. \tag{9}$$

where $m^+ > m^- \in \mathbb{R}$ are the thresholds above and below which an instance is considered good and bad respectively. Then, we fit a probabilistic classifier $\pi : \mathbb{R}^k \to [0,1]^3$ on $\mathcal{S}_{\boldsymbol{\theta}}^{\text{clf}}$, taking a vector of sorted scores $s(\mathbf{z}) \in \mathbb{R}^k$ as input and returning the estimated probabilities of $s(\mathbf{z})$ to belong to classes $-1$ (bad instances),

0 (average instances) and 1 (good instances). We finally define confidence function $u_{\mathrm{clf}}$ such that for a given unsorted vector of relevance scores $\mathbf{z}$, $u_{\mathrm{clf}}(\mathbf{z}) = \pi_2 \left( s(\mathbf{z}) \right) - \pi_0 \left( s(\mathbf{z}) \right)$ (probability that $\mathbf{z}$ stems from a good instance minus the probability that $\mathbf{z}$ stems from a bad instance). The higher $u_{\mathrm{clf}}(\mathbf{z})$, the more confident we are that $\mathbf{z} \in \mathbb{R}^k$ stems from a good instance.

In this work, we use a logistic regressor (Cox & Snell, 1989) with $l_2$ regularization parameter $\lambda = 0.1$, $u_{\mathrm{log}}$, and consider by default that the top and bottom 25% of instances with respect to metric function $m$ represent the good and bad instances respectively. We challenge this parameter in Appendix D.3.2.

### D.2.2  Distance-Based Approach

Confidence estimation can also be seen as a distance problem. While $u_{\mathrm{lin}}$ and $u_{\mathrm{log}}$ are fully supervised, it is possible to imagine a distance-based confidence function $u_{\mathrm{dist}}$ that evaluates how close an instance is to the set of good instances and how far it is from the set of bad instances.

For this purpose, based on thresholds $m^+$ and $m^-$ defined in Section D.2.1, we build $\mathcal{Z}_{\boldsymbol{\theta}}^+$ and $\mathcal{Z}_{\boldsymbol{\theta}}^-$, the sets composed of good and bad instances respectively. Formally,

$$\begin{cases} \mathcal{Z}_{\boldsymbol{\theta}}^+ = \left\{ \mathbf{z} : (\mathbf{z}, y) \in \mathcal{S}_{\boldsymbol{\theta}}, \ y > m^+ \right\} \\ \mathcal{Z}_{\boldsymbol{\theta}}^- = \left\{ \mathbf{z} : (\mathbf{z}, y) \in \mathcal{S}_{\boldsymbol{\theta}}, \ y < m^- \right\} \end{cases}.$$

Then, we construct a confidence scorer $u_{\mathrm{dist}}$ based on the distributions observed in $\mathcal{Z}_{\boldsymbol{\theta}}^+$ and $\mathcal{Z}_{\boldsymbol{\theta}}^-$. We define $u_{\mathrm{dist}} = \delta \left( s(\cdot), \mathcal{Z}_{\boldsymbol{\theta}}^- \right) - \delta \left( s(\cdot), \mathcal{Z}_{\boldsymbol{\theta}}^+ \right)$ as the difference between the distance to the set of bad instances and the distance to the set of good instances.[12] Intuitively, the higher $u_{\mathrm{dist}}(\mathbf{z})$, the more confident we are that $\mathbf{z}$ stems from a good instance.

In this experiment, we rely on the Mahalanobis distance (Mahalanobis, 2018; Lee et al., 2018; Ren et al., 2021). We compute the Mahalanobis distance between set $\mathcal{Z} \subset \mathbb{R}^k$ and $\mathbf{z} \in \mathbb{R}^k$ as

$$\delta_{\mathrm{mah}} \left( \mathbf{z}, \mathcal{Z} \right) = \left( \mathbf{z} - \boldsymbol{\mu}_{\mathcal{Z}} \right) \Sigma_{\mathcal{Z}}^{-1} \left( \mathbf{z} - \boldsymbol{\mu}_{\mathcal{Z}} \right)^T ,$$

where $\boldsymbol{\mu}_{\mathcal{Z}} \in \mathbb{R}^k$ is the average vector of reference set $\mathcal{Z}$ and $\Sigma_{\mathcal{Z}}^{-1} \in \mathbb{R}^{k \times k}$ is the inverse covariance matrix.

### D.3  Experimental Results

### D.3.1  Abstention Performance

Table 12 shows the normalized AUCs averaged model-wise for each dataset of the benchmark. We globally see that $u_{\mathrm{lin}}$ remains the best reference-based method, although $u_{\mathrm{log}}$ and $u_{\mathrm{mah}}$ both show satisfactory results, outperforming the reference-free baselines.

### D.3.2  Impact of Instance Qualification

In this section, we analyze the impact of the instance qualification threshold on the performance of $u_{\mathrm{log}}$ and $u_{\mathrm{mah}}$ confidence functions. More precisely, the idea of this experiment is to vary the qualification threshold[13] and to test its impact on nAUC over the whole benchmark. For the sake of readability, we present the results for the `ember-v1` model and for the mAP only (see Figure 7).

The first observation we can make is that the confidence functions $u_{\mathrm{log}}$ and $u_{\mathrm{mah}}$ are sensitive to variations in the qualification threshold. For all the datasets presented, it is noticed that both methods performs better for thresholds between 30% and 40%. Intuitively, too low a threshold would lead to considering too few instances as good or bad, making it difficult to provide proper characterization and therefore resulting in

---

[12]As for the other reference-based $u$ functions, $u_{\mathrm{dist}}$ takes a vector of unsorted scores as input, sorting being encompassed inside the function.

[13]As a reminder, a threshold of 10% means that good instances represent the top-10% with regard to the metric of interest, while bad instances represent the bottom-10%.

Table 12: nAUCs (in %) averaged over the whole set of available models, for every method including $u_{\log}$ and $u_{\mathrm{mah}}$.

| Dataset | Method Metric | $u_{\max}$ | $u_{\mathrm{std}}$ | $u_{\mathrm{1-2}}$ | $u_{\mathrm{lin}}$ | $u_{\mathrm{mlp}}$ | $u_{\mathrm{rf}}$ | $u_{\log}$ | $u_{\mathrm{mah}}$ |
|---|---|---|---|---|---|---|---|---|---|
| SciDocs | mAP | 49.2 | 58.7 | 4.7 | 65.4 | 65.6 | 67.3 | 68.8 | 65.8 |
|  | mNDCG | 54.9 | 62.9 | 10.2 | 66.6 | 65.3 | 68.6 | 70.2 | 66.8 |
|  | mRR | 80.2 | 80.3 | 41.3 | 80.5 | 76.8 | 70.7 | 78.3 | 65.6 |
| AskUbuntu | mAP | 24.1 | 16.2 | 7.3 | 22.0 | 19.4 | 14.1 | 19.9 | 10.1 |
|  | mNDCG | 28.1 | 15.7 | 8.6 | 24.5 | 21.2 | 14.0 | 22.0 | 10.5 |
|  | mRR | 32.7 | 16.6 | 14.0 | 26.2 | 22.8 | 14.8 | 24.1 | 11.3 |
| StackOverflow | mAP | 20.7 | 24.6 | 35.7 | 42.6 | 39.5 | 37.1 | 42.0 | 37.5 |
|  | mNDCG | 21.0 | 25.0 | 35.6 | 42.6 | 36.0 | 37.1 | 42.1 | 37.4 |
|  | mRR | 21.4 | 24.8 | 36.1 | 42.8 | 39.6 | 37.3 | 42.2 | 37.7 |
| Alloprof | mAP | 16.6 | 17.0 | 19.3 | 29.7 | 25.6 | 24.3 | 29.3 | 25.1 |
|  | mNDCG | 16.8 | 16.9 | 18.8 | 29.3 | 23.7 | 24.1 | 28.9 | 24.9 |
|  | mRR | 16.9 | 16.7 | 19.1 | 28.9 | 24.9 | 23.5 | 28.6 | 24.3 |
| CMedQAv1 | mAP | 24.4 | 23.3 | 22.3 | 30.6 | 23.3 | 24.8 | 29.0 | 25.7 |
|  | mNDCG | 25.0 | 23.5 | 22.3 | 30.7 | 22.2 | 25.4 | 27.0 | 24.8 |
|  | mRR | 26.5 | 24.2 | 23.4 | 31.5 | 24.7 | 25.5 | 30.2 | 25.5 |
| Mmarco | mAP | 30.1 | 31.0 | 39.4 | 34.0 | 32.0 | 29.7 | 25.1 | 22.7 |
|  | mNDCG | 30.5 | 30.9 | 38.8 | 34.1 | 30.3 | 28.7 | 25.1 | 23.3 |
|  | mRR | 30.1 | 31.0 | 39.6 | 34.3 | 31.8 | 29.4 | 23.7 | 22.8 |
| **Average** | mAP | 27.5 | 28.5 | 21.5 | 37.4 | 34.2 | 32.9 | 35.7 | 31.1 |
|  | mNDCG | 29.4 | 29.1 | 22.4 | 38.0 | 33.1 | 33.0 | 35.9 | 31.3 |
|  | mRR | 34.6 | 32.3 | 28.9 | 40.7 | 36.8 | 33.5 | 37.9 | 31.2 |

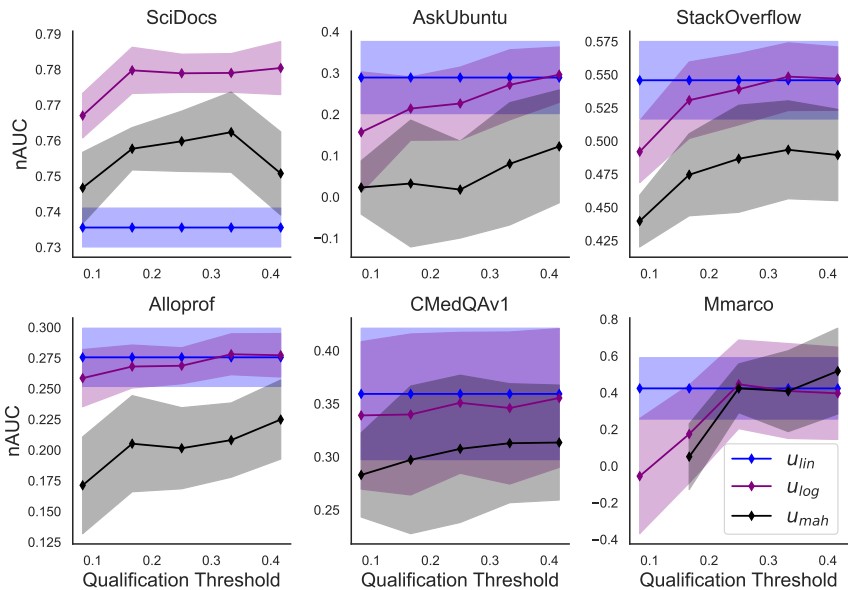

Figure 7: nAUC vs. instance quantile threshold for all datasets (model: `ember-v1`, metric: mAP).

less relevant confidence estimation. Conversely, a higher threshold creates less specific classes but with more instances, which seems to be better suited to our use case.

Additionally, while $u_{\mathrm{mah}}$ performs less well than $u_{\mathrm{lin}}$ overall, $u_{\log}$ competes fairly well when its qualification threshold is optimized. But the slight transformation required for the $\mathcal{S}_{\boldsymbol{\theta}}$ dataset (Equation 9) makes it a little less competitive in terms of computation time.

