# OpenReview forum: "Towards Trustworthy Reranking: A Simple yet Effective Abstention Mechanism"
_TMLR — Accepted by TMLR_

### Review · Reviewer_i7z9 · 2024-04-15

**Summary Of Contributions:**

This work proposes an abstention mechanism for neural information retrieval (NIR) and the corresponding evaluation metric. Two kinds of uncertainty functions are proposed to add the abstention mechanism, namely *reference-free* and *data-driven*. According to the empirical evaluation in Section 5, the proposed methods, especially $U_{\rm lin}$, can achieve good performance-abstention curves.

**Audience:**

Yes

**Broader Impact Concerns:**

The limitations are discussed.

**Claims And Evidence:**

Yes

**Requested Changes:**

# Requested Changes
- As Figure 1 is an illustration of the proposed method, consider moving it to earlier pages.

**Strengths And Weaknesses:**

## Strength
Although the abstention mechanism (also referred to as selective prediction) has been widely studied, this work is among the first to introduce such a mechanism to information retrieval.
- The proposed methods, both reference-free and data-driven, do not induce much extra computation.
## Weakness
Considering that the reference-based (also referred to as data-drive) method utilizes a separate model, the zero incurred cost claim is a bit overstated.
- The literature on *selective prediction* is absent.
- The finding that abstention performance is correlated with ranker performance is not surprising as better rankers can compute better-calibrated relevance scores. This actually induces another concern: the performance of the proposed method highly depends on the ranker performance, making it hard to select the threshold $\tau$ in practical settings. For instance, if the ranker is good, a low abstention rate could obtain good performance, while with a bad ranker, even a high abstention rate could underperform.
## Question
- How is reranking in this work different from retrieval?
- What metric $m$ is used? Do the findings hold for other metrics?
- Why is the $\tilde{u}$ curve in Figure 2 constant?

---

> ### Author Response · Authors · 2024-06-13
>
> We thank the reviewer for their detailed feedback. Below, we provide some responses as well as suggestions for modifications.
>
> **Weaknesses**
>
> * **The literature on selective prediction is absent.**
>
> We agree that the field of selective prediction is central to our work. We actually discussed number of works explicitly referring to it in Section 2.3, especially in the third paragraph (Chow, 1957; Bartlett & Wegkamp, 2008; El-Yaniv & Wiener, 2010; Cortes et al., 2016; Geifman & El-Yaniv, 2017; Geifman & El-Yaniv, 2019; Xin et al., 2021). Additionally, the fourth and fifth paragraphs mention white-box and black-box strategies for abstaining in cases of high uncertainty, which also relate to this notion. However, we realized that we did not name “selective prediction” explicitly, which is a mistake of ours and makes the reviewer’s comment perfectly justified. We therefore corrected this defect in the revised version of the manuscript.
>
> * **The finding that abstention performance is correlated with ranker performance is not surprising as better rankers can compute better-calibrated relevance scores. This actually induces another concern: the performance of the proposed method highly depends on the ranker performance, making it hard to select the threshold tau in practical settings. For instance, if the ranker is good, a low abstention rate could obtain good performance, while with a bad ranker, even a high abstention rate could underperform.**
>
> We agree with the reviewer’s point; it is not surprising to observe such a correlation. The initial idea of this section was to encourage potential users of our abstention mechanism not to apply it to models with excessively low raw performances. Indeed, the goal of our abstention mechanisms is not to transform a poor ranker into a good one, but rather to improve the quality of rankers that already show decent performance. However, as the reviewer points out, such an observation might seem to question the validity of abstention, at least as we presented it in our first version of the article. That is why we have proposed a slight reformulation of section 5.2 in order to convey our message more clearly.
>
> **Questions**
>
> * **How is reranking in this work different from retrieval?**
>
> We chose to focus our approach on the reranking phase because it is at the end of the chain in the most recent pipelines of information retrieval and RAG, which use the “retrieve then rerank” paradigm (Zhu et al., 2023), and we believed it made more sense to apply confidence estimation at the end of the chain rather than on an intermediate stage. Furthermore, the retrieval phase is performed on a  large scale database of documents (potentially hundreds of thousands or more), while the reranking phase focuses only on the top-K (K = 10, 50, 100) retrieved documents. Considering only the top retrieved documents allows for a more efficient estimation of our confidence functions. However, we agree it would be interesting in the future to work on generalizing our work to the retrieval case.
>
> * **What metric $m$ is used? Do the findings hold for other metrics?**
>
> $m$ is the generic term for the ranking metric used to assess the quality of a score vector with respect to a ground truth (see Section 2.1). In Section 4.2, we present the three ranking metrics used in our experiments: Average Precision, Normalized Discounted Cumulative Gain, and Reciprocal Rank. These three metrics are the most commonly used to evaluate the quality of a ranking in the field of information retrieval (Mitra et al., 2017; Muennighoff et al., 2022; Zhu et al., 2023). Furthermore, Table 1 presents the performance of our abstention strategies (normalised Area Under the Curve in %) for these three metrics of interest and show good performances for each of them.
>
> * **Why is the $\tilde{u}$ curve in Figure 2 constant?**
>
> As a reminder, $\tilde{u}$ is the confidence function associated with the $\tilde{\rho}$ mechanism, presented in Section 4.3. $\tilde{u}$ yields random abstention, i.e., ineffective abstention. A random (ineffective) abstention method cannot differentiate between good and bad instances. Thus, in expectation, a random abstention on the test dataset will result, in expectation again, in a constant performance whatever the abstention rate. We proposed some additional intuition about that in the revised version of our manuscript (section 4.3).
>
> **Requested changes**
>
> * **As Figure 1 is an illustration of the proposed method, consider moving it to earlier pages.**
>
> We have moved Figure 1 up to page 2.

---

### Review · Reviewer_vRAv · 2024-05-03

**Summary Of Contributions:**

This paper presents a simple but effective abstention method for reranking for neural information retrieval system with realistic industrial application constraints. Two methods to compute the confidence scores are proposed, one is for reference free situation which uses various ad hoc functions , the other is for data driven situation which uses linear regression, classification or distance functions. The experiments are carefully designed and extensive conducted to show the effectiveness of the proposed approach.

**Audience:**

Yes

**Claims And Evidence:**

Yes

**Requested Changes:**

Some experiments using MLP or other neural nets instead of  the linear regression function should be conducted.

**Strengths And Weaknesses:**

Strengths:

The paper is well written and easy to understand.

The motivation is clearly defined and the methods are quite simple and easy to implement.

The experimental setup is carefully designed and experimental results are convincing.

Weakness:

The proposed method is really simple and I'm wondering whether more sophisticated functions such as MLP neural nets will give better results than the linear regression function. Some experiments should be conducted.

---

> ### Author Response · Authors · 2024-06-13
>
> We appreciate the reviewer’s thorough feedback. We below address their remarks.
>
> **Weaknesses**
>
> * **The proposed method is really simple and I'm wondering whether more sophisticated functions such as MLP neural nets will give better results than the linear regression function. Some experiments should be conducted.**
>
> The idea behind our paper was to present a number of simple heuristics (both reference-free and reference-based) to assess the confidence level of document ranking models and motivate the application of abstention mechanisms to retrieval. We chose not to delve into more complex function classes partly because we wanted our mechanisms to work even in setups with limited labelled data. Nevertheless, we agree that adding a less trivial confidence function than a simple linear regression can provide interesting insights. In Appendix D, we added two regression-based confidence functions: $u_{\text{mlp}}$ (MLP with one hidden layer of size 128 and ReLU activation) and $u_{\text{rf}}$ (random forest with 100 estimators and squared error impurity criterion) to broaden our scope of experiments. But these two functions turned out to yield less convincing results than the simple linear regression, probably because the datasets of interest are not big enough for neural networks or random forests to present a real advantage over lighter machine learning techniques.
>
> **Requested changes**
>
> * **Some experiments using MLP or other neural nets instead of the linear regression function should be conducted.**
>
> As precised above, we incorporated experiments using an MLP and a random forest to the revised version of our manuscript (see Appendix D).

---

### Review · Reviewer_7qdD · 2024-06-10

**Summary Of Contributions:**

In this paper, authors address the problem of defining a strategy for abstention within the context of information retrieval. Additionally, to increase the practical applicability of their setup, they propose to work in a setting where the retrieval model might be a black-box/API, which does not allow the option to fine-tune or retrain the model. As no prior work exists in this particular setup, authors contribute in two aspects - i) proposing an algorithm to solve the problem and ii) define a benchmark which can be used to evaluate the proposed method.

**Audience:**

Yes

**Broader Impact Concerns:**

N/A.

**Claims And Evidence:**

Yes

**Requested Changes:**

As mentioned above, the paper overall is nicely written, has practical applications and can be useful. However, the core contribution - the algorithm does not have enough novelty. At this point, I do not have a lot of suggestions to provide that does not involve changing the core algorithm or adding some non-trivial technical contributions - which would then fundamentally change the paper.

**Strengths And Weaknesses:**

* The biggest strength of this paper is that it is practically applicable and can be quite useful to study for various real-world applications - especially with different text/multi-modal embeddings model providers that are being built.
* The proposed method (both reference-free and data-drive) are quite simple, intuitive and easy to follow and implement.
* The benchmark that was built for this paper can also be quite useful to study the overall abstention property, both with black-box and white-box embedding/retrieval algorithms.


* The main weakness of this paper is that the proposed method does not have enough novelty. There is novelty and practicality in the topic that this paper explores and also how the benchmark was created - but the core method feels like somewhat of a trivial baseline.

---

> ### Author Response · Authors · 2024-06-13
>
> We thank the reviewer for their comments. We are happy to see that they found our paper clear and well-written.
>
> **Weaknesses**
>
> * **The main weakness of this paper is that the proposed method does not have enough novelty. There is novelty and practicality in the topic that this paper explores and also how the benchmark was created - but the core method feels like somewhat of a trivial baseline.**
>
> As the reviewer rightly points it out, our contribution is not to propose a novel algorithm but rather to explore a new experimental area. In our article, we introduce the concept of selective prediction in the field of Information Retrieval through: 1) defining a realistic setting, 2) creating a comprehensive evaluation benchmark, and 3) presenting simple baselines (both reference-free and data-driven) that prove the applicability of abstention strategies in IR. Without resorting to overly complex approaches, we aim to show that the query-document similarity scores contain relevant calibration information when aggregated at the instance level, which is not immediately obvious.
>
> Besides, we would like to highlight the significant influence that various baseline-setting papers in the field of confidence estimation have had on the scientific community. A notable example is the work by Hendrycks & Gimpel (2016), which intuitively employs the maximum softmax probability to detect anomalous samples. This pioneering baseline has paved the way for further impactful contributions, such as those by Liang et al. (2017), Lee et al. (2018), and Hendrycks et al. (2018).
>
> This being said, to go beyond the heuristics we use in the paper, we conducted experiments with slightly more complex models in the revised version of the manuscript (see appendix D). While these strategies remain conceptually simple, they can theoretically capture finer patterns in the data than a linear regression. We however obtained mixed results with them, which led us to keep the linear regression as the data-driven baseline we include in the main body of the paper.

---

### Author Response · Authors · 2024-06-13

We would like to thank the reviewers for their detailed and very insightful feedback. We are super excited to see that our paper has sparked their interest, particularly regarding the novelty introduced by abstention in the field of retrieval, the clarity of our argument, and our experimental design. Overall, the reviewers felt it could be beneficial to make the literature review and experimental setup more comprehensive, issues that we have addressed in the revised version of our manuscript (in red font). We have also addressed more minor observations by incorporating some clarifications, both in the manuscript and below in direct response to their comments.

---

### Decision · Action_Editor_YyYn · 2024-08-26

**Recommendation:** Accept as is

**Comment:**

The paper describes and studies empirically methods for abstention in reranking in the context of information retrieval. Specifically, the authors consider two settings: reference free and data driven. In the reference free scenario, the authors do not assume access to any labels on the relevance of documents to a query, and the proposed methods are heuristics based on the predicted relevance scores. In the  data driven scenario, a linear mapping (or other parametric mappings) can be learned to convert the predicted relevance scores to a confidence score for abstention.

The methods discussed in the paper are simple, but the authors perform a thorough empirical study, and also propose a new benchmark. The authors also addressed the substantial comments of the reviewers in the manuscript, and the only remaining criticism of the paper is limited novelty, which is not required for acceptance to TMLR.

**Audience:**

The paper will be of interest to the information retrieval, RAG communities. The TMLR audience will be interested in the findings of this paper.

**Claims And Evidence:**

The claims in the paper are clearly stated and well-supported by empirical results. Section 5 of the paper is structured with specific statements and the corresponding empirical support.